# Templated synthesis of cubic crystalline single networks having large open-space lattices by polymer cubosomes

Yunju La[1], Jeongeun Song[1], Moon Gon Jeong[1,2], Arah Cho[1,2], Seon-Mi Jin[3,4], Eunji Lee[4] & Kyoung Taek Kim [1]

The synthesis of biophotonic crystals of insects, cubic crystalline single networks of chitin having large open-space lattices, requires the selective diffusion of monomers into only one of two non-intersecting water-channel networks embedded within the template, ordered smooth endoplasmic reticulum (OSER). Here we show that the topology of the circumferential bilayer of polymer cubosomes (PCs)—polymeric analogues to lipid cubic membranes and complex biological membranes—differentiate between two non-intersecting pore networks embedded in the cubic mesophase by sealing one network at the interface. Consequently, single networks having large lattice parameters (>240 nm) are synthesized by cross-linking of inorganic precursors within the open network of the PCs. Our results pave the way to create triply periodic structures of open-space lattices as photonic crystals and meta-materials without relying on complex multi-step fabrication. Our results also suggest a possible answer for how biophotonic single cubic networks are created, using OSER as templates.

[1] Department of Chemistry, Seoul National University, Seoul 08826, South Korea. [2] Department of Chemistry, Ulsan National Institute of Science and Technology (UNIST), Ulsan 44919, South Korea. [3] Graduate School of Analytical Science and Technology, Chungnam National University, Daejeon 34134, South Korea. [4] School of Materials Science and Engineering, Gwangju Institute of Science and Technology, Gwangju 61005, South Korea. These authors contributed equally: Yunju La, Jeongeun Song. Correspondence and requests for materials should be addressed to E.L. (email: eunjilee@gist.ac.kr) or to K.T.K. (email: ktkim72@snu.ac.kr)

Cubic crystalline single networks of large open-space lattices are ubiquitous in nature, from biophotonic crystals such as the wing scales of butterflies to the cuticles of weevils and beetles[1–6]. These structures, in particular, a single gyroid and diamond network, have been identified as promising metamaterials and photonic crystals exhibiting large omnidirectional optical bandgaps[7–10]. The biosynthesis of single networks of open-space cubic lattices such as simple cubic, single gyroid, and single diamond symmetries utilizes water-channel networks residing within the ordered smooth endoplasmic reticulum (OSER) of epithelial cells as templates for the polymerization of chitin[11,12]. OSER could be described as triply periodic minimal surfaces (TPMSs) composed of lipid bilayers[13], in which two nonintersecting water-channel networks are weaved in a long-range cubic crystalline order. However, how these complex biological membranes having two water-channel networks serve as a template for the formation of cubic crystalline single networks has not yet been clearly understood.

The use of ordered porous materials as sacrificial templates could be a facile pathway to synthesize cubic crystalline network structures of desired lattice and periodicity[14,15]. However, bicontinuous mesophases of surfactants and block copolymers (BCPs) internalize two identical channel networks arising from the thermodynamic preference of self-assembly to form structures of increased symmetry[16,17]. As a result, bicontinuously ordered templates yielded the formation of double network structures, of which the increased symmetry hampers the modulation of the property of propagating electromagnetic waves[18–21].

By employing calculations, cryogenic transmission electron microscopy (cryo-TEM) tomography, and atomic force microscopy (AFM), recent studies of the structural details of lipid cubic membranes[22] and their colloidally stabilized particles (cubosomes)[23] composed of TPMSs of lipid bilayers suggested that the outermost bilayer of the finitely-sized lipid cubic mesophases would adopt a topology in which one channel network is sealed while the other remains open to the surroundings[24–27]. This topology of the circumferential bilayer presumably arises due to the minimization of the interfacial energy by not revealing the hydrophobic compartment of the bilayer to the aqueous medium. Although this topological feature at the interface of lipid cubic membranes and cubosomes might offer the distinctive access toward embedded water-channel networks for external molecules[28], these synthetic analogues of the OSER have rarely been used as templates for the synthesis of single networks[29], partly due to their lattice parameter (<15 nm) being far smaller than the wavelength of visible light[30], combined with the physical and chemical fragility of lipid bilayers.

Polymer cubosome (PC)—colloidal particles composed of well-defined inverse bicontinuous cubic mesophases of BCP bilayers—possess cubic crystalline structures identical to those of lipid cubic membranes and complex biological membranes[31,32]. Compared to their lipid counterparts, PCs are structurally more robust under physical and chemical stresses, and their periodicity and pore size, which are an order of magnitude larger than those of lipid cubosomes, can be increased with increasing molecular weights of the polymer blocks constituting the BCPs.

In this article, we show that PCs share their interfacial topology with lipid cubic mesophases, which provides the required selectivity for diffusion of external molecules to embedded water channel networks, and consequently, serve as templates for the synthesis of single cubic networks with large open-space lattices that have been long-pursued as photonic crystals having complete photonic band gaps and metamaterials.

## Results

**Solution self-assembly of branched-linear BCPs into PCs.** In this study, we investigated the structures of PCs formed by the solution self-assembly of the branched-linear diblock copolymers composed of a hydrophilic tri-arm poly(ethylene glycol) and a hydrophobic polystyrene, $PEG550_3$-$PS_n$ ($n$ refers to the number average degree of polymerization of the PS; Fig. 1a)[33]. The PCs of $PEG550_3$-$PS_{150}$ (number-average molecular weight ($M_n$) = 15.9 kDa, polydispersity index ($Đ$) = 1.04, weight fraction of the PEG chains ($f_{PEG}$) = 10.6%) and $PEG550_3$-$PS_{168}$ ($M_n$ = 17.8 kDa, $Đ$ = 1.06, $f_{PEG}$ = 9.4%) were chosen as model systems, which represent two distinct lattices comprising bicontinuous cubic mesophases (Fig. 1b).

Small-angle X-ray scattering (SAXS) results and scanning electron microscopy (SEM) images of the PCs of $PEG550_3$-$PS_{150}$ (average diameter ($d$) determined by analysis of SEM images = 11.1 ± 6.7 μm; ± indicates range of values) confirmed that the internal bicontinuous cubic structure of the BCP bilayer had symmetry corresponding to the Schwarz P surface ($Im\bar{3}m$ space group) with the lattice parameter ($a$) of 60.7 nm. The internal cubic mesophase of the PCs of $PEG550_3$-$PS_{168}$ ($d$ = 15.9 ± 7.9 μm) was assigned to the Schwarz D surface of BCP bilayers ($Pn\bar{3}m$ space group, $a$ = 45.2 nm) (Supplementary Figs. 1 and 2).

In contrast to the interfacial morphology of lipid cubosomes[23,24], PCs have an interface composed of a perforated bilayer shell surrounding the internal TPMS of BCP bilayers without disordered vesicular outer layers (Fig. 1c, e, Supplementary Fig. 2a and b). This structural characteristic allowed us to directly visualize the detailed morphology of the interface of the PC. By comparing SEM images of the interface and the internal mesophase of the PC, we inferred that the density of the surface pores was approximately a half the pore density observed from the internal mesophase (Fig. 1d, f, Supplementary Fig. 2c–f).

As shown in Fig. 1g, the computer-generated (100) plane of the Schwarz P surface was superimposed on the SEM image of the surface of the PC of $PEG550_3$-$PS_{150}$, indicating that, at the interface, only one simple cubic network embedded in the Schwarz P surface remained open (indicated by a magenta square), leaving the other channel network closed (indicated by a green square). In contrast, the internal TPMS residing under the circumferential bilayer showed two sets of simple cubic channel networks (Fig. 1h). The distance between adjacent pores was 65 ± 3 nm, and the diagonal distance of the tetragonal pores was 90 ± 4 nm (indicated by a white arrow in Fig. 1g). The fully open lattice of the Schwarz P surface ($a$ = 60.7 nm) residing in the PC showed the adjacent and diagonal pore distances of 60.7 and 42.9 nm, respectively (Fig. 1h).

With the superimposed (111) plane of the lattice of the Schwarz D surface, analyzing SEM images of the perforated lamellar shell of the PCs of $PEG550_3$-$PS_{168}$ also revealed the presence of the open (indicated by a magenta diamond) and closed (indicated by a green diamond) channels at the surface of the PC (Fig. 1i). The distance between the pores at the surface of the PCs consisting of the Schwarz D surface was 71 ± 5 nm, which corresponded to the $\sqrt{2}a$, the diagonal distance of the pores of the (111) plane of the Schwarz D surface ($a$ = 45.2 nm) with one open channel[25]. In contrast, the SEM image of the internal (111) plane of the Schwarz D surface clearly indicated the presence of two open channel networks (Fig. 1j). These results showed that the interface of the PC obscured one of two internal water-channel networks, as suggested by computational studies of lipid cubosomes[26,27].

We performed TEM tomography of a PC to illustrate the connectivity of the surface pores to the internal channel networks embedded within the BCP cubic mesophases (Supplementary Movie 1). TEM images of a PC were taken at different tilt angles, offering successive recording of two-dimensional (2-D) projections from different views. The tilt series of images were carefully aligned and used to reconstruct the entire volume of a PC with nanometer-scale spatial resolution using back-projection

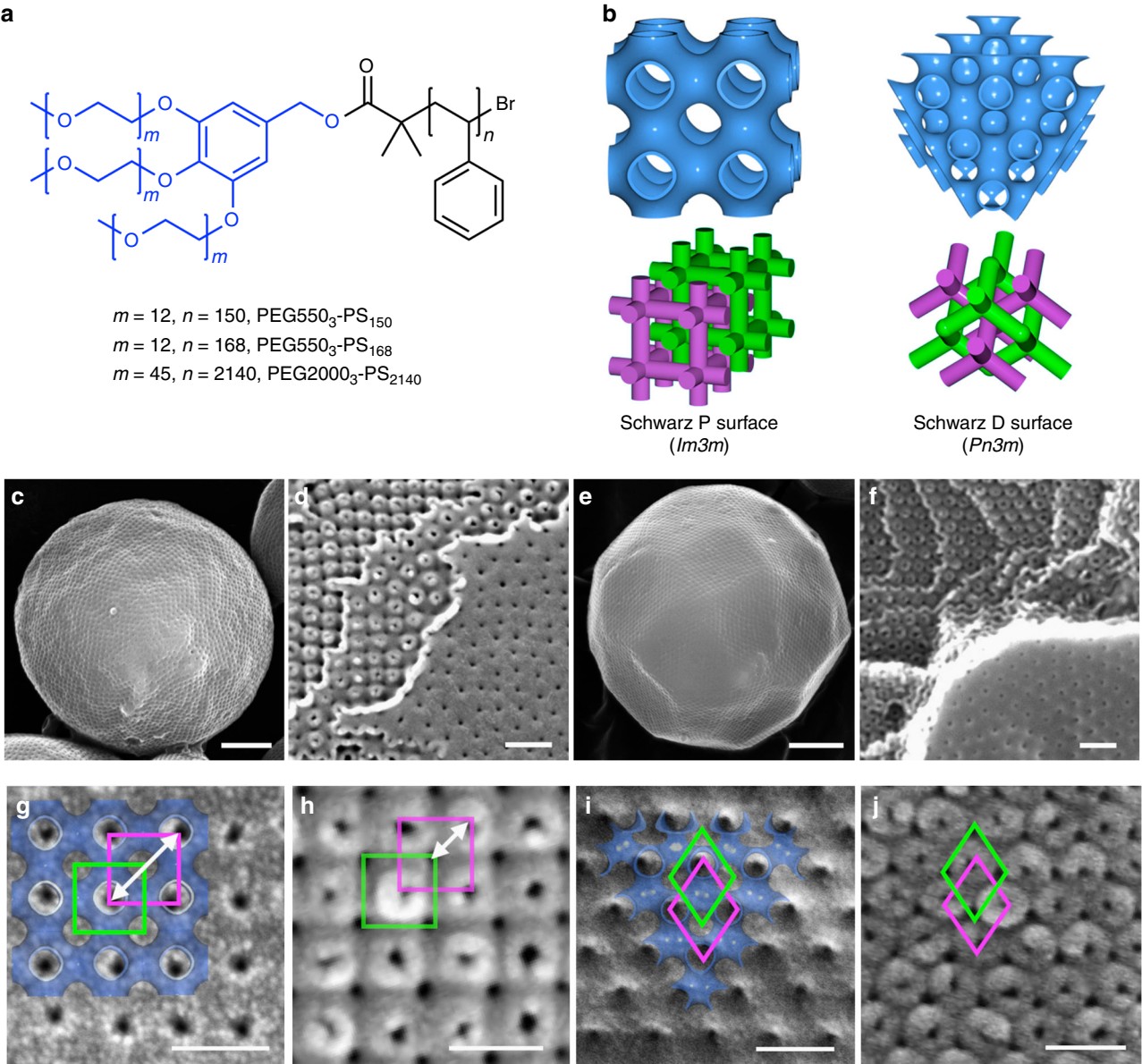

**Fig. 1** Interfacial topology of polymer cubosomes (PCs). **a** Chemical structure of the BCP used in this study. **b** Two representative lattices of the internal cubic mesophases of BCP bilayers. Color-coded skeletal networks represent two non-intersecting channel networks embedded in the BCP cubic mesophases. **c**, **e** SEM images showing the spherical morphology of the PCs of PEG550$_3$-PS$_{150}$ (**c**) and PEG550$_3$-PS$_{168}$ (**e**). Scale bars are 1 μm. **d**, **f** SEM images of fractured PCs of PEG550$_3$-PS$_{150}$ (**d**) and PEG550$_3$-PS$_{168}$ (**f**) showing the interface and internal cubic mesophase of the BCP bilayers. Scale bars are 200 nm. **g**–**j** SEM images showing the surface topology of the PC of PEG550$_3$-PS$_{150}$ (*Im$\bar{3}$m* space group) (**g**) and PEG550$_3$-PS$_{168}$ (*Pn$\bar{3}$m* space group) (**i**). The superimposed images are computer-generated (100) plane of the Schwarz P surface (**g**) and (111) plane of the Schwarz D surface (**i**). SEM images of the internal Schwarz P (**h**) and D (**j**) surfaces of the corresponding PCs indicate the interfacial capping of one of two internal channel networks. Scale bars are 100 nm

algorithms in IMOD. The three-dimensionally (3-D) reconstructed volume image of the PC of PEG550$_3$-PS$_{150}$, shown in Fig. 2a, established the spherical morphology with perforated surface bilayer surrounding the internal mesophase. A 3-D reconstructed volume image of TEM of the PC of PEG550$_3$-PS$_{168}$ revealed similar morphological characteristics (Supplementary Fig. 3). The TEM tomograms of the internal cubic mesophases of these PCs showed the presence of internal lattices of *Im$\bar{3}$m* and *Pn$\bar{3}$m* space groups (Supplementary Fig. 4).

We mapped the TEM tomogram of the PCs by reversing the phase of the reconstructed tomogram of the PC of PEG550$_3$-PS$_{150}$ (Fig. 2b, c). The open- and closed-channel networks (color-coded

magenta for open; green for closed) exhibited triply periodic connectivity within the PC. The cross-sectional views (with each intervals of 12 nm) of the TEM tomograms of the PC consisting of Schwarz P surface along the z-axis showed the presence of two non-intersecting cubic channel networks embedded in the PC (Fig. 2d).

Because two cubic channel networks embedded within a PC are non-intersecting, the closure of one channel network by the circumferential bilayer of a PC would create a closed compartment[24,28]. To confirm this, we self-assembled PEG550$_3$-PS$_{150}$ in the presence of fluorescein sodium salt in water (1 mM) to form PCs (Fig. 2e). After dialysis against water to remove all

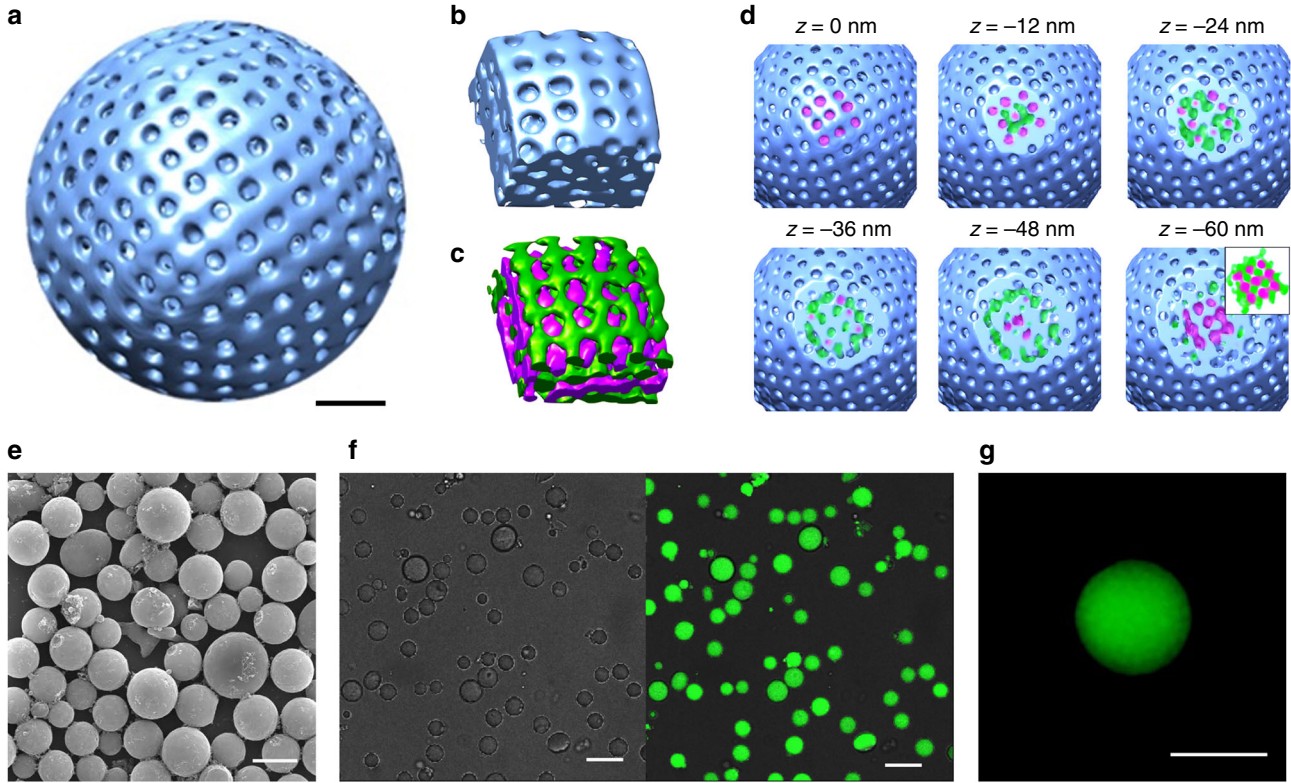

**Fig. 2** Three-dimensional representation of the interface and internal structures of PCs. **a** 3-D reconstructed TEM tomograms of the PC of PEG550₃-PS₁₅₀. Scale bar is 100 nm. **b** The 3-D tomogram of the part of the PC of PEG550₃-PS₁₅₀ showing the circumferential bilayer and internal cubic membrane. **c** The internal non-intersecting water channel networks embedded within the PC. **d** Cross-sectional images of the 3-D tomogram along z direction. The inset shows merged images of double networks. The magenta channel is open at the interface, and the green remains closed. **e** SEM image of the PCs of PEG550₃-PS₁₅₀ encapsulating fluorescein within the closed channel networks. Scale bar is 10 μm. **f** Optical and confocal laser scanning microscopy of the PCs. Scale bars are 10 μm. **g** SIM image of the fluorescein-encapsulating PCs of PEG550₃-PS₁₅₀. Scale bar is 2 μm

organic solvents and excess dyes, confocal laser scanning microscopy (CLSM) and structured illumination microscopy (SIM) revealed fluorescence ($\lambda_{em} = 515$ nm) contained within the PCs (Fig. 2f, g and Supplementary Fig. 5). SEM and SAXS analysis of the PCs self-assembled in the presence of fluorescein showed the same surface topology and well-defined internal structures of double networks (Supplementary Fig. 5h and i). These results indicate that the fluorescein is entrapped in a closed water channel network without being diffused to the open channel network of PCs connected to the surrounding medium. For the control experiment, we mixed pre-formed PCs with a fluorescein solution. Diffusion of fluorescein into the open channel of the pre-formed PCs did not show any retention of dye molecules after purification (Supplementary Fig. 5e–g). These results suggest that only open channel network of the PC would be available for free diffusion of external molecules in solution.

**Templated synthesis of cubic crystalline networks**. Based on our structural findings, we tested the accessibility of the open channel network of the PCs by backfilling them with the solution of tetraethyl orthosilicate (TEOS) to replicate the internal channel network as an inorganic skeletal framework by sol–gel reaction. After cross-linking under acidic conditions, the solvent-soluble templates were removed by immersing the particles in tetrahydrofuran (THF) to avoid isotropic shrinkage of the silica network during calcination. SEM images of the silica replica of the PCs showed spherical morphology with well-defined internal networks reflecting the shape of the PC used as a template, suggesting the cross-linking of TEOS in the vicinity of PEG corona of the bilayers (Fig. 3a, b, and Supplementary Fig. 6).

The SAXS results from the skeletal silica networks of the PCs of PEG550₃-PS₁₅₀ indicated that the lattice of the skeletal structure was transformed from primitive cubic double network ($Im\bar{3}m$ space group) to a simple cubic network ($Pm\bar{3}m$ space group) (Fig. 3c). The lattice parameter (a) of the simple cubic network was 56.5 nm, which was nearly identical to the value of the PC templates (having $Im\bar{3}m$ symmetry of $a = 60.7$ nm). The SAXS peaks of the silica replica of the PCs of PEG550₃-PS₁₆₈ were assigned to a single diamond lattice ($Fd\bar{3}m$ space group) with the lattice parameter of 90.7 nm, a two-fold increase from the value (45.2 nm) of the double diamond lattice ($Pn\bar{3}m$ space group) of the PC template (Fig. 3d). The estimation of the scattering correlation length ($\xi \approx 2\pi/\Delta q$, where $\Delta q$ is the full-width at half maximum of the first diffraction peak) of the PCs and the resulting replica (Supplementary Fig. 1)[34] suggested that the entire open-channel networks were replicated into cubic crystalline frameworks of silica without disrupting the crystalline order.

SEM and TEM images of the fractured particles replicated from the PCs of PEG550₃-PS₁₅₀ revealed the cubic network having six-fold junctions, of which the structural symmetry was inherited from the primitive cubic lattice of the channel networks embedded in the PC template (Fig. 3e). The silica replica of the channel network embedded in the PC of PEG550₃-PS₁₆₈ was also identified by SEM and TEM, which showed the skeletal frameworks as having the nodes of four-fold symmetry (Fig. 3f). These results indicated that the resulting structures were indeed single networks of primitive cubic and diamond lattices rather than the double network structures that were overlapped by azimuthal shifting upon removal of the template[19–21,35]. Recently, Akbar and coworkers reported the platinum nanowire networks

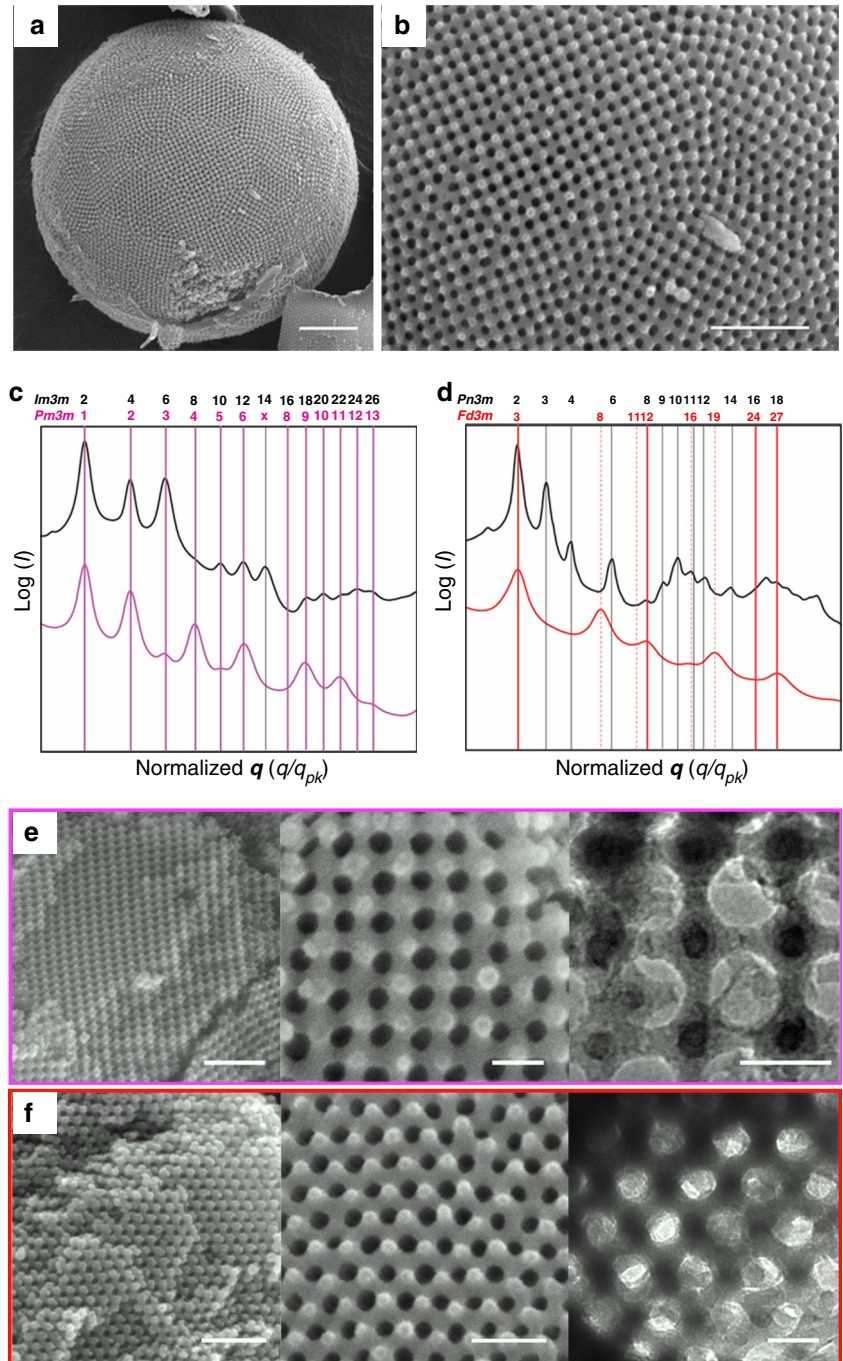

**Fig. 3** Structural characterization of silica single networks templated by PCs. **a, b** SEM images of the silica replica templated by PCs of PEG550$_3$-PS$_{150}$ showing spherical morphology (**a**) and porous surface (**b**). Scale bars are 1 μm (**a**) and 500 nm (**b**). **c, d** SAXS results of skeletal silica replica (color lines) synthesized from the PC templates (black lines). The symmetry transition from the corresponding double networks of the PCs ($Im\bar{3}m$ for **c** and $Pn\bar{3}m$ for **d**) to simple cubic ($Pm\bar{3}m$, $a = 56.5$ nm) (**c**) and single diamond ($Fd\bar{3}m$, $a = 90.7$ nm) (**d**). Vertical lines correspond to the expected Bragg peak positional ratios (magenta and red; common peaks, dashed red; $Fd\bar{3}m$ only, black; PCs only). **e, f** SEM and TEM images of the single cubic silica network synthesized from the PCs of PEG550$_3$-PS$_{150}$ (**e**) and the single diamond silica network obtained from the PCs of PEG550$_3$-PS$_{168}$ templates (**f**). Scale bars are 300 nm (left), 100 nm (middle), and 50 nm (right)

having a single diamond symmetry with a lattice parameter of 13.2 nm via electrodeposition of metal precursors by using a double diamond cubic phase of lipids as a template[29]. Our results indicated that the cubosomes composed of BCPs possessed the identical structures to those of lipid cubic mesophases and could serve as templates for the synthesis of single cubic networks with larger lattice parameters[36].

**Synthesis of single networks having large open-space lattices.** The single cubic network of inorganic materials having high dielectric contrast could develop omnidirectional photonic band gap if the lattice dimension is commensurate with the wavelength of the incident light[37,38]. As the dimension of the polymer chain scales with the square-root of its molecular weight, the lattice parameter of the BCP cubic mesophases should be

proportionally enlarged by increasing the molecular weights of the polymer blocks constituting the BCP[39]. Therefore, we synthesized high molecular weight BCP, PEG2000$_3$-PS$_{2140}$ ($M_n$ = 197 kDa, $Đ$ = 1.19, $f_{PEG}$ = 2.7%) with an anticipation of the increased lattice parameter of the internal TPMS comprising the PCs. The dioxane solution of PEG2000$_3$-PS$_{2140}$ (1 wt%) was allowed to self-assemble into PCs in a saturated humidity chamber, in which the solvent evaporates while water diffuses into the solution[40]. This method ensured an introduction of a poor solvent (water) into a BCP solution in an extended period, which allowed self-assembly of very high molecular weight BCPs

without being trapped in non-equilibrium structures. SEM and TEM images of the resulting PCs showed the presence of internal cubic mesophases (Fig. 4a–c). The diameter of PCs is relatively small ($d$ = 3.3 ± 2.2 μm), indicating the increased colloidal stability arising from the presence of long PEG chains on the surface of the bilayers[33]. The distance between pores at the surface of the resulting PCs, measured by SEM and TEM, was 270 ± 30 nm. The SAXS results of the PC showed broad peaks suggesting significant distortion of the lattice in the internal cubic mesophases of the PCs (Supplementary Fig. 7). From the primary peak of the SAXS result, the lattice parameter

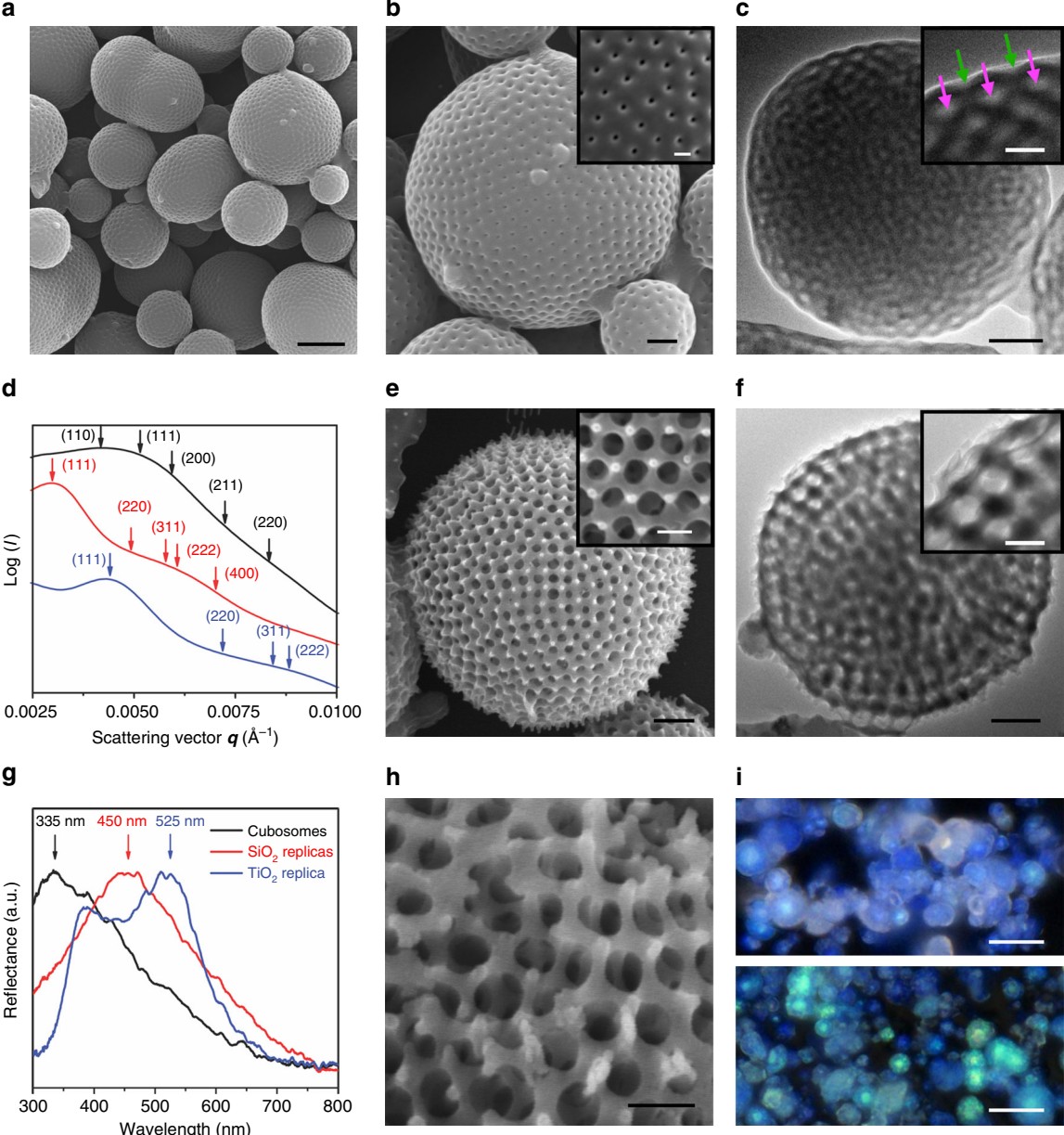

**Fig. 4** PCs and cubic crystalline single networks having large lattice parameters. **a**, **b** SEM images of PCs of PEG2000$_3$-PS$_{2140}$ display the large pore distance at the surface. Scale bars are 2 μm (**a**), 500 nm and 200 nm (**b** and inset). **c** TEM image of the PC exhibited the distance of two pores at the surface is 240 nm. The open and closed channels are marked by green and magenta arrows, respectively. Scale bars are 500 nm and 200 nm (the inset). **d** SAXS results of the PCs (black line, assigned by assuming $Pn\overline{3}m$ symmetry, $a$ = 213 nm) and single networks of silica (red, assigned to $Fd\overline{3}m$, $a$ = 361 nm) and titania (blue, assigned to $Fd\overline{3}m$, $a$ = 247 nm). **e**, **f** SEM and TEM images of silica single cubic networks templated by PCs of PEG2000$_3$-PS$_{2140}$. Scale bars are 500 nm and 200 nm (insets). **g** UV–vis reflectance spectrometry of the PCs (black), silica networks (red), and titania networks (blue). **h** SEM image of fractured titania single cubic network showing internal $Fd\overline{3}m$ symmetry. Scale bar is 200 nm. **i** Optical microscope images of silica (top) and titania (bottom) networks. Scale bars are 10 μm

was estimated to be 213 nm by assuming the internal Schwarz D surface ($Pn\bar{3}m$ space group).

The replication of the internal single diamond channel network with TEOS and titanium (IV) isopropoxide (Ti(OPr)$_4$) was performed by sol–gel reactions under acidic conditions. The resulting skeletal frameworks were characterized by SEM, TEM, and SAXS, which indicated the presence of cubic crystalline single network (Fig. 4d–f, h, Supplementary Fig. 7). SAXS and SEM images of the SiO$_2$ and TiO$_2$ replica of the PC of PEG2000$_3$-PS$_{2140}$ suggested that the lattice of the single cubic network mainly retain a single diamond structure while significant distortion of the lattice was present in the skeletal network (Supplementary Fig. 8). We tried to assign the SAXS results obtained from these skeletal networks, which was unsuccessful due to the significant broadness of the peaks (Supplementary Fig. 7g). In spite of distortion of the lattices partly caused by the high molecular weight of the BCP and the high curvature at the surface of the PCs, the replicated SiO$_2$ and TiO$_2$ structures showed single cubic networks having the nodes of four-fold symmetry. By assuming $Fd\bar{3}m$ space group, SAXS results showed that the lattice parameter of the silica network was 361 nm, while the value of the TiO$_2$ network decreased to 247 nm due to the shrinkage of TiO$_2$ during calcination (Fig. 4d). The particles of cubic networks of silica displayed a blue hue centered at 450 nm on UV–vis reflection spectroscopy as well as optical microscopy. The calcined single diamond networks of TiO$_2$ also showed iridescence, arising from the optical bandgap centered at 525 nm measured by UV–vis reflection spectroscopy. The peak near 400 nm indicates an intrinsic absorption edge of TiO$_2$ (Fig. 4g, i). Our results suggest that photonic crystals composed of single cubic networks of open-space cubic lattices could be synthesized from cross-linking of molecular precursors using PCs as templates.

## The inversion of interfacial topology of PCs by fusion.

For the synthesis of biophotonic single networks, the OSER should be exposed to the extracellular space by fusion with the plasma membrane for allowing the diffusion of chitin to the open channel network of the template. We propose a possible mechanism for fusion of the plasma membrane and OSER, which should lead to the inversion of the topology[41] of the OSER at the newly formed interface (i.e., the open channel turns to the closed channel, and the closed channel opens only at the interface created by fusion). The interfacial topology of the cubic membrane within the cell should remain unchanged. This topological inversion at the interface created by fusion would allow diffusion of external monomers only into the open channel created by the fusion process, while retaining the cytoplasmic content within a cell (Supplementary Fig. 9).

To support this mechanism, we fused polymersomes of PEG550$_3$-PS$_{120}$ encapsulating fluorescein sodium salt in the inner compartment (1 mM) (Supplementary Fig. 10a–c) and PCs of PEG550$_3$-PS$_{150}$ in a mixture of 1,4-dioxane and water (1:1 v/v). In this condition, polymersomes of PEG550$_3$-PS$_{120}$ could undergo fusion although the population of the fused polymersomes was small (Supplementary Fig. 10d). After removal of excess polymersomes by centrifugation, we observed both unfused free polymersomes and fused PCs showing the presence of encapsulated fluorescein (Fig. 5a, b). SEM and TEM images of PCs after fusion showed deformed polymersomes protruding from the surface of the PC (Fig. 5c, d). SIM and CLSM images of the fused PCs, only observed in a low population, clearly indicated that the fluorescein originating from the inner lumen of the polymersomes diffused into the closed compartment of the PC, which only opened at the interface created by the fusion process (Fig. 5 and Supplementary Fig. 10e and 11). The diffused fluorescein molecules were retained within the closed channel of the PCs after dialysis of the dispersion against water, indicating that the

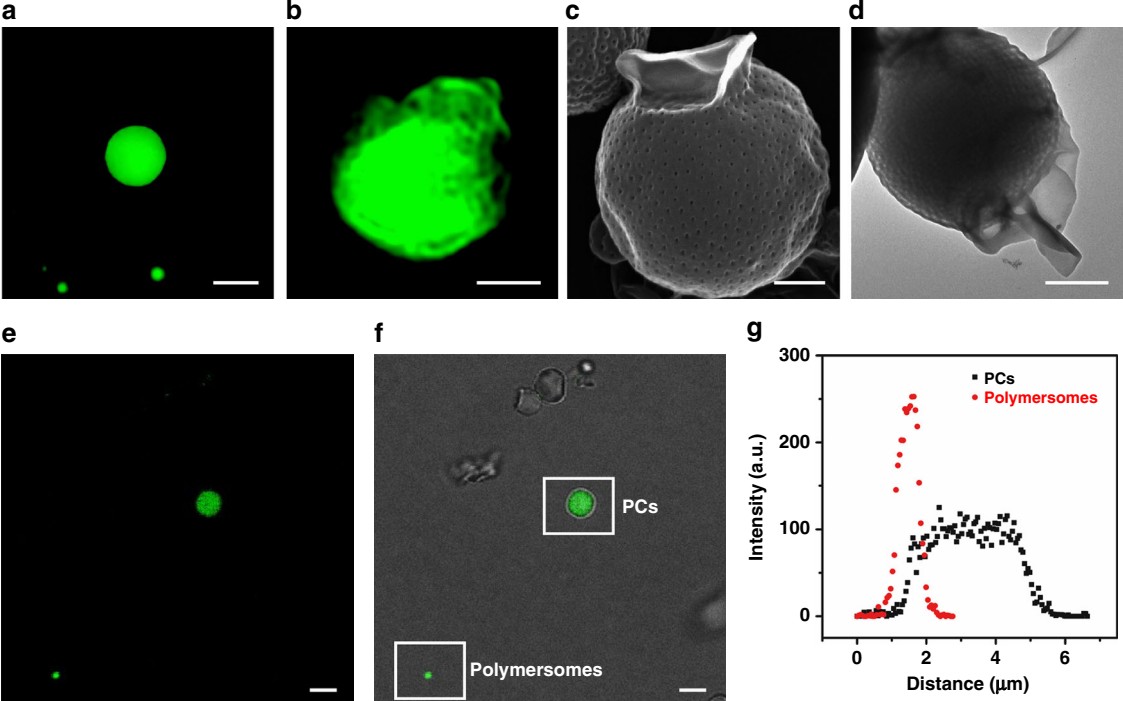

**Fig. 5** Fusion of fluorescein-encapsulating polymersomes and PCs. **a** SIM image showing the fusion product of polymersome/PC and free polymersomes. Scale bars are 2 μm. **b** SIM image of fused PC showing deformed polymersomes tethered to the surface of the PC. Scale bar is 1 μm. **c, d** SEM (**c**) and TEM (**d**) image showing the fusion of fluorescein-encapsulating polymersomes and PCs. Scale bars are 500 nm. **e, f** CLSM images showing both fluorescein-encapsulating polymersomes and PCs. The merged image (**f**) shows that only PCs fused with polymersomes retain fluorescence. Scale bars are 5 μm. **g** The Intensity profile of CLSM images of fluorescein-encapsulating polymersomes and PCs revealing the distribution of fluorescein

topology of the surface of the PC was only reversed at the interface created by fusion with polymersomes (Fig. 5g).

One of the proposed mechanisms for the synthesis of photonic crystals of insects using biological cubic membranes as the templates is the pre-folding model: the intracellular SER membrane forms a cubosome-like phase (OSER). This pre-folding of the SER into the OSER requires a process, by which one of two non-intersecting water channel networks embedded within the OSER is connected to the extracellular space. After the polymerization of chitin by diffusion of the monomers into the channel networks exposed to the extracellular space, the cell demises, and only a single network structure is left as scales[6]. Our suggested mechanism could complement the proposed mechanism (pre-folding model) by suggesting the possible means to expose one water channel network to the extracellular space, which allowed the influx of chitin into the OSER template residing in the cell.

## Discussion

In summary, we synthesized cubic crystalline single networks of inorganic oxides having large open-space lattices using PCs as templates. Sharing the crystallographic and structural characteristics with lipid cubosomes and cubic membranes, PCs, finite-sized particles of cubic mesophases of BCP bilayers, possess the interfacial bilayers of BCPs, of which the topology obscuring one of two non-intersecting internal water-channel networks to become a closed compartment, while the other remains as the open channel network that can be used as templates for cross-linking of molecular precursors. The presence of the resulting closed compartment of a PC was confirmed by the retention of fluorescent molecules within the closed channel, indicating that the topological distinction between two internal non-intersecting water-channel networks should provide the selective diffusion of external molecules into the open channel of the PC. We demonstrated the synthesis of cubic crystalline single networks of primitive cubic and single diamond lattices by utilizing this selective diffusion of inorganic precursors to the open cubic channel networks and consequent sol–gel reaction. Utilizing the proportionality of the lattice dimension and the molecular weight of the BCPs that self-assemble to form PCs, cubic crystalline single networks of silica and titania with large lattice parameter (>240 nm) are synthesized, which display structural colors analogous to the iridescence exhibited by biophotonic crystals of insects. Utilizing the topological selection at the interfacial bilayer of cubic mesophases of BCPs, our demonstration paves the way to synthesize photonic bandgap materials and metamaterials in a fashion inspired by the synthesis of biophotonic crystals of arthropod. Thanks to the physical and chemical stability of PC templates, we envisage that a wide variety of materials such as polymers, metals, and inorganics could be used to form cubic crystalline networks, which may find applications in metamaterials and photonic bandgap materials. In addition, our results suggest a possible answer to understand how biological single networks can be synthesized by using bicontinuously ordered complex membranes.

## Methods

**Synthesis of block copolymers**. BCPs, PEG550$_3$-PS$_n$ and PEG2000$_3$-PS$_n$, were synthesized by the atom-transfer radical polymerization of styrene in the presence of the corresponding macroinitiators. Detailed synthetic procedures were reported previously[33].

**Preparation of PCs**. A representative procedure is described. 20 mg of PEG550$_3$-PS$_{150}$ was charged in a vial equipped with a magnetic stirring bar. 2 mL of dioxane was added to this vial. The polymer solution in the vial capped with a rubber septum was stirred for 2 h at room temperature. To this solution, 2 mL of water was added via a syringe pump at a rate of 0.5 mL h$^{-1}$ with stirring (600 rpm). After addition of water, the suspension was subjected to dialysis against water by using a dialysis membrane (MW cut-off 12–14 kDa) with frequent change of water for 24

h. For PEG2000$_3$-PS$_{2140}$, 10 mg of the BCP was dissolved in 1,4-dioxane (990 mg) in a 4 mL vial and stirred for 3 h at room temperature. A humidity chamber was prepared by mixing 20 mL of 1,4-dioxane and 20 mL of water in a 100 mL vial including a cylindrical column to put a glass substrate. 0.4 mL of the polymer solution was cast on the 2 × 2 cm$^2$ glass substrate on the column in the humidity chamber. The humidity chamber was then sealed. After 4 h, the glass substrate was immersed into excess water. The PCs were collected after removing 1,4-dioxane by solvent exchange in water.

**Encapsulation of fluorescent molecules in PCs**. PCs of PEG550$_3$-PS$_{150}$ were prepared by using a fluorescein sodium salt solution (1 mM in water). After dialysis of the suspension against water for 24 h, the suspension was further purified by the repeated filtration on a centrifugal filter (Amicon, MW cut-off 100 kDa) and exchange of the medium with pure water. This procedure was repeated until the filtrate showed no sign of the presence of free fluorescein on a fluorometer. As a control experiment, the concentrated PCs of PEG550$_3$-PS$_{150}$ were mixed with a fluorescein solution (1 mM). The resulting mixture was equilibrated under gentle shaking at room temperature for 24 h. This mixture was purified by repeated filtrations on a centrifugal filter and exchange of the medium.

**Encapsulation of fluorescent dyes by fusion of polymersomes and PCs**. Polymersomes of PEG550$_3$-PS$_{120}$ encapsulating fluorescein sodium salt within the lumen were prepared by self-assembling the BCP in dioxane with water (1 mM fluorescein sodium salt) by following the procedure mentioned above. After dialysis, the suspension was further purified by size-exclusion chromatography (Sephadex G-100) to remove residual fluorescence dye by using water as an eluent. The concentrated solution of the polymersomes was added to the suspension of PCs of PEG550$_3$-PS$_{150}$ (PC:polymersome = 1:5 v/v). This mixture was dialyzed against a dioxane:water (1:1 v/v) for 24 h. The dialyzed mixture was incubated at room temperature for 2 weeks. For fluorescence and electron microscopy, the polymersome/PC mixture was centrifuged (1000 rpm) in a methanol/water mixture (1:1 v/v) to remove free polymersomes. After centrifugation, the medium was switched to water by centrifugation.

**Templated synthesis of silica and titania networks using PCs**. TEOS sol was prepared by mixing TEOS, ethanol, water, and HCl (molar ratios = 1:3:1.5 × 10$^{-5}$) at room temperature for 12 h. PCs were placed on the filter paper. After addition of a drop of TEOS solution to the PCs, the backfilled PCs were quickly placed on a fresh filter paper. The resulting sol-impregnated PCs were placed in a humidity chamber filled with 5 M HCl vapor. After 12 h, the polymer template was removed by dissolution of BCPs in THF. Titanium (IV) isopropoxide sol was prepared by slowly adding 2 mL of titanium isopropoxide to a mixture of acid (1.6 mL of trifluoroacetic acid and 0.4 mL of hydrochloric acid). After 20 min, 4 mL of iso-propanol was added to this mixture. The backfilled PCs with titania sol were prepared in the same method as the preparation of silica replicas of PCs. The resulting sol-impregnated cubosome particles were allowed to dry for 24 h. The dried sample was calcined at 500 °C under air for 6 h.

## Data availability

The data that support the findings of this work are available from the authors on reasonable request. See author contributions for specific data sets.

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

## Acknowledgements

The authors thank Prof. Hyunhyub Ko and Ms. Ayoung Choe for measuring the reflectance spectroscopy. The authors also thank Dr. Hyungju Ahn for SAXS measurement at PAL. This work was supported by National Science Foundation (NSF) of Korea (2016R1A2B3015089 and 2016R1A2B4012322) and Seoul National University (SNU) for the support by Creative-Pioneering Researchers Program.

## Author contributions

K.T.K. conceived and designed the experiments. Y.L., J.S., A.C., and M.G.J. carried out most synthetic works and characterization. A.C. conducted encapsulation and fusion experiments. S.-M.J. and E.L. performed and analyzed TEM tomography. K.T.K. wrote the initial manuscript. All authors commented and contributed on the initial and revised manuscript.

## Additional information

**Competing interests:** The authors declare no competing interests.

