## [Peer Review File · Nature Communications]

Reviewers' comments:

Reviewer #1 (Remarks to the Author):

Paper contains many original findings, and is of interest for many fields such as physics, chemistry, materials science and biology.

The findings include the following:

- Synthesis of polymer cubosomes having bicontinuous cubic structures. To my knowledge, It is for the first time demonstrated that polymer cubosomes have one water channel close and one open to the outside.
- Synthesis of single network replica with very large lattice parameters using polymer cubosomes, which confirms that one network is open and the other one is close. This single domain is similar to what is found in insects.
- Fusion of polymersomes with polymer cubosomes, which suggests a possible mechanism in biology for the fusion of the plasma membrane with the endoplasmic reticulum and how chitin single network can be formed.

Findings are based on a crystallographic analysis, which is in general very well done and explained.

Therefore, I recommend it to be published in Nature Communications after careful considerations of the following comments.

- Methods section is incomplete. In particular details on preparation of samples for TEM, SEM & confocal & SAXS and on electron tomography, reflectance spectrometry and instruments used are absent
- P.6 lines 122-129. As far as I am concerned, there is no demonstration that one water channel is open and that the other one is close in this section. With the present text and figures, it can only be inferred that at least some water inside the polymer cubosomes is isolated from the outside. Therefore, conclusions for this part should be milder or additional data should be provided. Note that with the evidences obtained from electron microscopy and from the synthesis of replica, it is obvious that one channel is open to the outside and one is close.
- P.6-8 lines 131- 163. The experimental part is lengthy and conclusion from the data (and literature) is obvious. It can be shorten and details can be put in supplementary data. It is also not completely clear what is original on this part since it is not the first time that replicas are made using bicontinuous cubic structure as templates, e.g. see Akbar, S., Elliott, J.M., Rittman, M. & Squires, A.M. Facile production of ordered 3D platinum nanowire networks with "single diamond" bicontinuous cubic morphology. *Advanced Materials* 25, 1160-1164 (2013). This aspect on previous work is briefly mentioned in the introduction, but it should be discussed here in light of the obtained results together with the originality of the present work (e.g. first time that geometry similar to what is found in insect has been synthesized and what are the biological implications?).
- P.8-lines 181-185. Here, the text does not reflect figures and figure captions. In the text, it is claimed that replica correspond to a mixtures of Pm-3m and Fd-3m while in Fig.4 only Fd-3m is indicated. What is supplementary fig.7 adding more, why are the SAXS data of Fig.4 and supplementary fig.7 so different? In addition, in SAXS data of Fig.4 and supplementary fig.7, peaks are difficult to identify. A mixture of structure might explain this finding, but an expert in SAXS should back-up this explanation and maybe simulation of the SAXS data obtained by mixing the 2 structures should be performed. SAXS is crucial in the paper, it is a little bit surprising that the scientist, who performed the SAXS is not a co-author (and therefore likely not part of the interpretation).
- P.10 Lines 214-216 or p.11 lines 239-240. I think one needs to explain the parallel between the finding of the present work and biology (formation of biophotonic crystals in insect), starting from the formation of bicontinuous structures in the endoplasmic reticulum, then going to the fusion of membrane with the bicontinuous structure and chitin diffusion. This also asks a question: what happens with the initial bicontinuous structures in biophotonic crystals: is it still there or is it

removed (e.g. chemically, by diffusion....)?

Reviewer #2 (Remarks to the Author):

The manuscript describes the use of cubosomes for templating inorganic mesophases and their corresponding optical properties. Critically, the authors have identified, using careful imaging, that one of the two pore networks is closed to the surrounding, and could exploit this behavior to carefully show that one pore network can be filled selectively. This kind of "surface reconstruction" leading to closure of one pore of an ordered lattice in cubosomes, and moreover the use of it, is to the best of my knowledge unprecedented. The quality and depth of the study is very high and I recommend publication of the article after addressing the following issues.

1. Defect mapping: it is very astonishing to realize that one network is fully closed considering that the surface of the quasi-spherical cubosomes must be defect-free. Can you provide further quantification of possible defects at edges of rather flat surfaces of the cubosomes (e.g. Figure 1c,d).
2. Figure 2d: z-sections: Why are the outer pores in the z-planes not color coded? Are they impossible to assign? And why is this the case?
3. The vesicle fusion experiments are very clever. I enjoyed this section very much. However, the fluorescence profile in Figure 6g are somewhat misleading. The profile of the vesicle is diffraction limited. It implies a maximum in the center, but only because the flat profile cannot be resolved at higher magnifications if looking at the z-section. I would recommend to remove this. It is unclear what the authors wish to demonstrate with this?
4. Control experiment: I am missing the following control experiment. Strictly speaking the fluorescence section does not prove that only one channel is filled, as the microscopy does not allow to resolve these details. Imagine there was a chemical interaction of what so ever origin so that fluorescein segregates to the PEO or to the PEO-PS interface, then very similar images could be obtained, even for open pores. Hence, I recommend the following controls: (i) mixtures of prepared cubosomes and fluorescein, separation via sephadex and imaging. Ideally the cubosomes should not be enriched with the fluorophore to claim entrapment in closed pores. (ii) The authors should encapsulate one fluorophore in the closed pore network and add another fluorophore to infiltrate the open pores. The cubosomes with the larger lattice spacing should allow a SIM based imaging should allow for a separation of both dyes in the orthogonal channels.
5. The relation to biosynthesis of insect cuticles is to my opinion a bit overstated. Do we really learn from this processes that the formation of the cuticles works similarly?
6. Figure 5 and 6 should be merged.
7. A recent minireview on cubosomes and similar structures should be cited "DOI: 10.1002/anie.201703765"

Reviewer #3 (Remarks to the Author):

See below.

Comments on “Templated synthesis of cubic crystalline single networks having large open-space lattices by polymer cubosomes” by La and coworkers

La and co-workers present a way to produce polymer cubosomes (PCs) and transform their internal structure into inorganic materials. The authors show that only one network of their PCs is connected to the environment and that this can be used to trap fluorescent molecules in the closed-off compartment or refill the dual network from the outside. While their results are certainly interesting and novel, some of the results need to be described better (and more honest) and to be placed into context. Let me detail these points below.

Structure of the PCs? The authors have synthesized various PEG-PS polymers to form PCs with either a D or P minimal surface internal morphology. While the structure of the small PCs is convincingly either D/P, it is not clear how the channel morphology. However, it is not clear how the pore size (or its filling fraction) is related to the polymer morphology. Are the network directly connected to the PS/PEO fraction?

Large PCs? What I fully disagree with is the structure determination of the large PEG2000-PS polymer of figure 4. Rather than wanting to ascribe this structure to a mixture of P/D, the results show that it is rather a disordered network of pores. The structure factors could also “just” be structure factors of pores and look quite similar to normal spheres. The SEM/TEM images do not convince me that the structure is indeed a single or a mix of morphologies. Also, the optical results underpin a more quasi-ordered network as the optics quite closely resembled the quasi-ordered network, e.g. found in birds (Tinbergen et al. JEB 2011 or Saranathan JRSI 2012). Please adjust accordingly.

Why is only one network connected to the outside? While the results of Figures 1-4 are really beautiful, the results and discussion of figure 5-6 are purely hypothetical. I would argue that Figure 5 is really clear and trivial and should go either to the SI or out, as it does not really make sense to me. Please make sure that this is pure speculation that it is connected to the growth of SERs as this is not convincingly shown, as you also do not discuss as to why only one channel of your PCs is connected. Please amend.

Figure layout. I would strongly suggest to group the results for the different PCs in the figures, e.g. move all belonging to Schwarz D to the right and the one's to P to the left. This will make it easier to compare. Also, many figures are missing scale bars (Fig. 1 G-J, Fig. 2 i). It also not clear which colored curve in Figure 3 belongs to which sample.

Language. Please have the language checked again as some “the” are too many, some “the” are missing. Furthermore. “single networks” of...? In particular, the summary needs some work as “obscuring” and “to become” are more teleological and scientifically correct. The summary could also profit from a bit more detail.

Methods. While the PC process is clear, the process has no details on the spectroscopic characterization, the SAXS measurements or the details on other instruments. Please add to allow reproducibility.

Detailed comments.

P. 2: define “low coordinate”

P. 4: don’t start with “in order to test our idea”, quite jargon

P. 5: the 3D reconstruction description needs to be improved.

P. 6: add note on volume fractions of the channels here.

P. 7: what is black/red in the curves? Add description here.

L. 163: a recent paper <https://onlinelibrary.wiley.com/doi/abs/10.1002/sml.201802328> highlights this in quite some detail. Maybe add a note to this ms here?

L. 175 and elsewhere: give errors where possible.

L. 190-195: as described above the optics is more similar to quasi-ordered networks than to a mix of two unit cells, please adjust.

L. 208: the description of Figure 6 needs to be improved. Not really clear.

L. 221: what is exactly the process of behind the fact that only one network is connected? Only with that you can draw further conclusions.

L. 233: how can a structural color be reminiscent of iridescent of single networks? Quite a stretch and linguistically wrong. Please rewrite.

We thank all reviewers for their comments and criticisms. We have revised the manuscript by carefully considering reviewers' comments and suggestions. The additions and corrections related to referee's comments are highlighted with a yellow marker in the revised manuscript. Figures were slightly modified for better visibility. Our responses to the comments raised by the reviewers are listed below.

REVIEWER #1

Paper contains many original findings, and is of interest for many fields such as physics, chemistry, materials science and biology. The findings include the following:

- Synthesis of polymer cubosomes having bicontinuous cubic structures. To my knowledge, It is for the first time demonstrated that polymer cubosomes have one water channel close and one open to the outside.
- Synthesis of single network replica with very large lattice parameters using polymer cubosomes, which confirms that one network is open and the other one is close. This single domain is similar to what is found in insects.
- Fusion of polymersomes with polymer cubosomes, which suggests a possible mechanism in biology for the fusion of the plasma membrane with the endoplasmic reticulum and how chitin single network can be formed.

Findings are based on a crystallographic analysis, which is in general very well done and explained. Therefore, I recommend it to be published in Nature Communications after careful considerations of the following comments.

(1) Methods section is incomplete. In particular details on preparation of samples for TEM, SEM & confocal & SAXS and on electron tomography, reflectance spectrometry and instruments used are absent

[Our response]

➔ As the referee commented, we found that one page was missing from the supplementary information about methods section. We corrected this in our new supplementary information. We apologize for this mistake.

(2) P.6 lines 122-129. As far as I am concerned, there is no demonstration that one water channel is open and that the other one is close in this section. With the present text and figures, it can only be inferred that at least some water inside the PCs is isolated from the outside. Therefore, conclusions for this part should be milder or additional data should be provided. Note that with the evidences obtained from electron microscopy and from the synthesis of replica, it is obvious that one channel is open to the outside and one is close.

[Our response]

➔ In this section, we intended to show the existence of one water-channel network as a closed compartment within the polymer cubosome (PC). The block copolymer was allowed to self-assemble into PCs in the presence of fluorescein in water. When the PC is formed, surface topology of the PC closes one water-channel network at the interfacial

bilayer. Because a PC internalizes two non-intersecting water channels, diffusion of molecules between two channels is not allowed. After the self-assembly, we removed unencapsulated fluorescent molecules by dialysis against water and repeated centrifugations and exchanges of a medium (our method section). After these purification steps, the confocal laser microscopy images of the PCs at different focal planes clearly showed that fluorescein was well-retained within the PC. (Supplementary Figure 5d). This indicates that the fluorescein is entrapped in a closed water-channel network without being diffused to the open channel network. In addition, SEM and SAXS analysis of the PCs self-assembled in the presence of fluorescein showed the presence of well-defined internal structures having bicontinuous water channel networks (Supplementary Figure 5h and i). We added this SEM and SAXS results of the PCs self-assembled in the presence of fluorescein to show the presence of well-defined bicontinuous cubic phases (P surface). For the control experiment, fluorescein was allowed to diffuse into the open channel network of the PCs formed without fluorescent molecules. We confirmed that the fluorescein diffused into an ‘open channel’ of PCs cannot be retained during the same purification process. We added the confocal microscopy images from this control experiment in the supplementary information (Supplementary Figure 5e-g). We believe that our results support the presence of one water channel network of the PC as a closed compartment.

➔ **Modified Supplementary Figure 5**

Supplementary Figure 5. CLSM and SIM images of the PCs of PEG550₃-PS₁₅₀. **a,b,c**, CLSM images of PCs self-assembled in the presence of Fluorescein in water ($\lambda_{\text{Ex}} = 460 \text{ nm}$, $\lambda_{\text{Em}} = 515 \text{ nm}$) : (a) dark field, (b) bright field, (c) merged. The merged image showed the presence of dye in the water channels in PCs. **d**, SIM images obtained from different focal planes (interval of z-direction : 100 nm). Scale bars : 2 μm . **e,f,g**, CLSM images of PCs mixed with fluorescein in water after purification : (e) dark field, (f) bright field, (g) merged. The dark field and merged images did not show any retention of dye molecules. **h**, SEM images of fluorescein-encapsulating PCs. The inset shows magnified view of perforated surface of colloidal PC. **i**, SAXS result of fluorescein-encapsulating PCs shows that the phase and lattice constant are similar to PCs formed without fluorescein ($Im\bar{3}m$, $a = 60.2 \text{ nm}$).

(3) P.6-8 lines 131- 163. The experimental part is lengthy and conclusion from the data (and literature) is obvious. It can be shorten and details can be put in supplementary data. It is also not completely clear what is original on this part since it is not the first time that replicas are made using bicontinuous cubic structure as templates, e.g. see Akbar, S., Elliott, J.M., Rittman, M. & Squires, A.M. Facile production of ordered 3D platinum nanowire networks with "single diamond" bicontinuous cubic morphology. *Advanced Materials* 25, 1160-1164 (2013). This aspect on previous work is briefly mentioned in the introduction, but it should be discussed here in light of the obtained results together with the originality of the present work (e.g. first time that geometry similar to what is found in insect has been synthesized and what are the biological implications?).

[Our response]

→ We thank the reviewer for insightful comments and for drawing our attention to the previous example of the synthesis of single diamond networks templated by lipid cubic mesophases. Akbar and coworkers reported the production of platinum nanowire networks with single diamond morphology. They demonstrated the electrodeposition of HCPA platinum metal precursor through the template having double diamond cubic phases. The lattice constant of their film is ca. 13 nm because the template was formed by self-assembly of lipid molecules. One of the motivations of our current work is to achieve large lattice dimensions of cubic mesophases, which could be found from biological photonic crystals of insects. The lattice constant of biophotonic crystals of insects is commensurate with a wavelength of visible light. We aimed to achieve this nature's length scale by using the self-assembly of block copolymers with a high molecular weight. We showed that the photonic band gap appeared when the lattice constant of the PC is increased to ~210 nm and the nature's length scale could be obtained by the replication of the open water channel network of the PC. With reviewer's comment in mind, we corrected our manuscript (P. 7-8 lines 163-168) to indicate the previous work by Akbar and coworkers and to deliver our goal to achieve large lattice dimensions by using PCs. To shorten the experimental discussion, we also moved the detailed description of scattering correlation lengths into the supplementary information (P. 7 lines 152-155, Supplementary Figure 1).

→ **Revised manuscript (P. 7-8 lines 163-168)**

Recently, Akbar and coworkers reported the platinum nanowire networks having a single diamond symmetry with a lattice parameter of 13.2 nm via electrodeposition of metal precursors by using a double diamond cubic phase of lipids as a template³⁶. Our results indicated that the PC composed of BCPs possessed the identical structures to those of lipid cubic mesophases and could serve as templates for the synthesis of single cubic networks with larger lattice parameters³⁷.

→ **Revised manuscript (P. 7 lines 152-155)**

The estimation of the scattering correlation length ($\xi \approx 2\pi/\Delta q$, where Δq is the full-width at half maximum of the first diffraction peak) of the PCs and the resulting replica (Supplementary Fig. 1)³³ suggested that the entire open-channel networks were replicated into cubic crystalline frameworks of silica without disrupting the crystalline order.

→ **Added figure legend in Supplementary Figure 1**

The size of crystallites was estimated as 673 nm for the internal Schwarz P surface of PEG550₃-PS₁₅₀ and 716 nm for its replica. For the Schwarz D surface of PEG550₃-PS₁₆₈, the size of crystallite was estimated as 833 nm and 516 nm for its single diamond replica.

(4) P.8-lines 181-185. Here, the text does not reflect figures and figure captions. In the text, it is claimed that replica correspond to a mixtures of Pm-3m and Fd-3m while in Fig.4 only Fd-3m is indicated. What is supplementary fig.7 adding more, why are the SAXS data of Fig.4 and supplementary fig.7 so different? In addition, in SAXS data of Fig.4 and supplementary fig.7, peaks are difficult to identify. A mixture of structure might explain this finding, but an expert in SAXS should back-up this explanation and maybe simulation of the SAXS data obtained by mixing the 2 structures should be performed. SAXS is crucial in the paper, it is a little bit surprising that the scientist, who performed the SAXS is not a co-author (and therefore likely not part of the interpretation).

[Our response]

→ We thank the reviewer for bringing up this important point. We agree with reviewer's concern with our original assignment of SAXS results. We reassigned our structures as a single diamond network with noticeable distortion of lattices especially at the surface of the particles. We give our detailed explanation for this correction below:

For the solution self-assembly of block copolymers having a PS hydrophobic block (a glassy polymer at ambient temperature), the co-solvent method (gradual introduction of water (a non-solvent toward PS) in the organic solvent (dioxane) dissolving the block copolymer) is used for the solubilization of the block copolymer. As the water content in the organic solvent increases, the hydrophobic block is destabilized and assembled to form aggregate. Essentially, the mobility of the glassy PS block is lost when the water content reaches to a critical value. After this point, the block copolymers became immobile, and the organic solvent is removed from the water by dialysis. In our case, water was slowly added to the dioxane solution of a block copolymer until the water content reaches to 50 vol. %. Under this condition, the self-assembled structures could be kinetically frozen especially when the rate of addition of water is fast and the molecular weight of the hydrophobic PS block is large.

In the case of the PCs of PEG550₃-PS₁₅₀ and PEG550₃-PS₁₆₈, the order of the internal structure was very high partly because the block copolymers could reach the thermodynamic equilibrium during the self-assembly. Also, the presence of relatively short PEG chains ($M_n = 550 \text{ g mol}^{-1}$) in the hydrophilic corona of the bilayer membrane provides limited colloidal stability to the resulting PCs. Therefore, the average diameter of the PCs is large ($> 11 \mu\text{m}$) compared to the lattice constant of the bicontinuous cubic phases. These facts contributed to the size of the internal crystalline domains of a PC to be large enough for showing clear diffraction peaks by SAXS experiments. For these samples, we observed well-resolved peaks from the SAXS experiments, which could be assigned to the corresponding lattices (Supplementary Figure 1).

However, for the PCs of PEG2000₃-PS₂₁₄₀, the mobility of the block copolymer during the self-assembly is limited due to the substantially higher molecular weight of the PS block compared to the previous block copolymers. The presence of long PEG chains on the surface of the bilayer should increase colloidal stability of the PCs (ref. 32), which resulted in the formation of relatively small particles (average diameter = $2.7 \mu\text{m}$). Due to the small diameter, the surface curvature of the PCs of PEG2000₃-PS₂₁₄₀ is much higher than the surface curvature of larger PCs of PEG550₃-PS₁₅₀ and PEG550₃-PS₁₆₈. This high

surface curvature might result in the distortion of the lattices at the surface of the PCs. Also, the limited mobility of the block copolymer makes the self-assembled structures difficult to reach the equilibrium during the self-assembly, which contributes to the increase of disorder in the internal bicontinuous mesophases of the PCs. Also, the small diameter of the PCs of PEG2000₃-PS₂₁₄₀ gave a background noise from particle scattering in the SAXS experiments, which render the peaks to be broaden significantly as shown in Figure 4d. We assigned this result to $Pn\bar{3}m$ symmetry by considering the SEM images of the PCs.

Consequently, SiO₂ and TiO₂ replica of the PCs showed broad peaks by SAXS experiments, which could be assigned to $Fd\bar{3}m$ symmetry as shown in Supplementary Figure 7. From the SEM and TEM studies of these skeletal replicas of the PCs, we observed that the lattices of the skeletal structures consisted mostly of a single diamond network ($Fd\bar{3}m$ symmetry, four-fold junctions, Figure 4e inset and 4h). However, we also observed the presence of primitive cubic networks from the surface of the skeletal replica, which might arise from the distortion of the lattice at the surface of the replica particles. Based on the analysis of SEM images, we inferred that a minor fraction of the networks had a primitive cubic ($Pm\bar{3}m$) structures, which mostly appeared at the surface of spherical particles composed of single networks (supplementary figure 8).

Based on these observations and our previous results of the PCs having a mixed phase (*Angew. Chem. Int. Ed.* **54**, 10483–10487 (2015); *ACS Nano* **9**, 3084–3096 (2015)), we assigned our SAXS results to the mixed phases. However, due to the less resolved and broad SAXS peaks, we could not confirm the coexistence of two phases within the PCs and replica particles. Therefore, we reassigned our SAXS data of replicas to $Fd\bar{3}m$ symmetry with noticeable disorder in the structures. The disorder of the lattices of the PC, however, did not disrupt the interfacial topology, resulting in the formation of single network structures. We revised our manuscript accordingly (P. 8 lines 180-182, 183-186, P.9 lines 189-197) and added revised data in the supplementary information.

The SAXS data in Supplementary Figure 7f is obtained by replacing the SAXS data of replicas in Figure 4d with the normalized q value in order to compare the relative peak positions (ref. 3). We modified the figure legend to avoid confusion.

→ **Revised manuscript (P. 8 lines 180-182)**

The diameter of PCs is relatively small ($d = 2.7 \mu\text{m}$), indicating the increased colloidal stability arising from the presence of long PEG chains on the surface of the bilayers³².

→ **Revised manuscript (P. 8 lines 183-186)**

The SAXS results of the PC showed broad peaks suggesting significant distortion of the lattice in the internal cubic mesophases of the PCs (Supplementary Fig 7). From the primary peak of the SAXS result, the lattice parameter was estimated to 213 nm by assuming that the internal Schwarz D surface ($Pn\bar{3}m$ space group).

→ **Revised manuscript (P. 9 lines 189-197)**

The resulting skeletal frameworks were characterized by SEM, TEM, and SAXS, which indicated the presence of cubic crystalline single network (Fig. 4d–f, h, Supplementary Fig. 7). SAXS and SEM images of the SiO₂ and TiO₂ replica of the PC of PEG2000₃-PS₂₁₄₀ suggested that the lattice of the single cubic network was mainly retain a single diamond structure while significant distortion of the lattice was present in the skeletal network (Supplementary Fig. 8). We tried to assign the SAXS results obtained from these skeletal networks, which was unsuccessful due to significant broadness of the peaks

(Supplementary Fig. 7g). In spite of distortion of the lattices partly caused by the high molecular weight of the BCP and the high curvature at the surface of the PCs, the replicated SiO_2 and TiO_2 structures showed a single cubic network having nodes of 4-fold symmetry.

→ **Modified Supplementary Figure 7**

Supplementary Figure 7. Structural characterization of the titania replica of PCs of PEG2000₃-PS₂₁₄₀. **a**, Low-magnification SEM image showing spherical replicas. **b,c**, SEM images of structures of the TiO_2 replicas showing $Pm\bar{3}m$ space group (b) and $Fd\bar{3}m$ space group (c). **d,e**, TEM images showing the single network of replicas. **f**, Normalized SAXS results of skeletal silica replicas (red line) and titania replicas (blue line). Vertical lines correspond to the expected Bragg peak positional ratios of $Fd\bar{3}m$ space group. **g**, SAXS results of the PCs and replicas assigned to mixed phases. The expected peak position of $Im\bar{3}m$ space group of PCs was calculated from the lattice parameter relationship of two phases ($a_{p\text{ surface}}/a_{d\text{ surface}} = 1.279$). Polymer cubosomes; $Pn\bar{3}m$ ($a = 213$ nm, black) and $Im\bar{3}m$ ($a = 273$ nm, red). SiO_2 replicas; $Fd\bar{3}m$ ($a = 361$ nm, red) and $Pm\bar{3}m$ ($a = 231$ nm, black). TiO_2 replicas; $Fd\bar{3}m$ ($a = 247$ nm, blue) and $Pm\bar{3}m$ ($a = 159$ nm, black).

→ Added Supplementary Figure 8

Supplementary Figure 8. SEM images of the SiO₂ replica of PCs of PEG2000₃-PS₂₁₄₀. **a,b**, Low-magnification SEM image showing the single cubic network of the spherical replica (a) and magnified view of surfaces of the replica showing distorted lattice (b). **c**, SEM image of the single cubic network of SiO₂ replicas showing $Pm\bar{3}m$ space group on the surfaces.

(5) P.10 Lines 214-216 or p.11 lines 239-240. I think one needs to explain the parallel between the finding of the present work and biology (formation of biophotonic crystals in insect), starting from the formation of bicontinuous structures in the endoplasmic reticulum, then going to the fusion of membrane with the bicontinuous structure and chitin diffusion. This also asks a question: what happens with the initial bicontinuous structures in biophotonic crystals: is it still there or is it removed (e.g. chemically, by diffusion...)?

[Our response]

→ There are two hypotheses on the mechanism for the synthesis of photonic crystals of insects using biological cubic membranes as the templates. One is the “prefolding” model: the intracellular SER membrane forms cubosome-like phases known as ordered smooth endoplasmic reticulum (OSER). From the folding of SER and plasma membranes, one of two non-intersecting water channel networks embedded within the OSER is connected to the extracellular space which allows the deposition of chitin. After the polymerization of chitin and cell death, only a single network structure is left as scales. The other is the “cofolding” model in which chitin extrusion and membrane folding occur simultaneously. (ref. 6, *J. Morphol.* **202**, 69–88 (1989)).

In our study, we showed that the interfacial topology of the PCs could force only one of two non-intersecting water-channel networks to be exposed to the surroundings. This surface topology could be reversed upon the fusion with a polymer bilayer, resulting in the switch of the accessibility of the channel networks (i.e., open to closed, closed to open) at the interface created by fusion. We assumed that if the same process could happen in the epithelial cell of insects, the smooth endoplasmic reticulum (OSER) forms ordered particulates (cubosomes), which subsequently fuse with a plasma membrane. The fusion between the cell membrane and the cubosome (OSER) exposes the formerly closed channel network to the extracellular space, while the previously open channel network turned to a closed channel residing inside the cell. Chitin diffuses and polymerizes only in the exposed channel networks, and the cytosolic content remains within the cell. Essentially, this epithelial cell is diseased, and only chitinous materials remain as a scale. (The structure of the scale is hierarchical.).

Our experiments provide a model system for the ‘pre-folding’ model and a possible mechanism suggesting how the OSER could connect only one water-channel to the extracellular space without losing the cytosolic contents, which has not been suggested yet. With reviewer’s comment in mind, we revised our manuscript (P. 10 lines 228-235).

→ **Revised manuscript (P. 10 lines 228-235)**

The proposed mechanism for the synthesis of photonic crystals of insects using biological cubic membranes as the templates is the “prefolding” model: the intracellular SER membrane forms a cubosome-like phase (OSER). From the folding of SER and plasma membranes, one of two non-intersecting water channel networks embedded within the OSER is connected to the extracellular space which allows the deposition of chitin. After the polymerization of chitin and cell death, only a single network structure is left as scales⁶. Our suggested mechanism could complement the proposed mechanism (prefolding model) by suggesting the possible means to expose one water channel network to the extracellular space, which allowed the influx of chitin into the OSER template residing in the cell.

REVIEWER #2

The manuscript describes the use of cubosomes for templating inorganic mesophases and their corresponding optical properties. Critically, the authors have identified, using careful imaging, that one of the two pore networks is closed to the surrounding, and could exploit this behavior to carefully show that one pre network can be filled selectively. This kind of “surface reconstruction” leading to closure of one pore of an ordered lattice in cubosomes, and moreover the use of it, is to the best of my knowledge unprecedented. The quality and depth of the study is very high and I recommend publication of the article after addressing the following issues.

(1) Defect mapping: it is very astonishing to realize that one network is fully closed considering that the surface of the quasi-spherical cubosomes must be defect-free. Can you provide further quantification of possible defects at edges of rather flat surfaces of the cubosomes (e.g. Figure 1c,d).

[Our response]

→ The surface morphology of Polymer cubosomes (PCs) is formed by minimizing the interfacial energy of the circumferential bilayer during self-assembly, by which the hydrophobic compartment of the bilayer is not revealed to the aqueous solution. This thermodynamic process creates virtually no defects in terms of the topology of the bilayer at the interface.

Based on our experiments, the topology of the surface (which channel is open or closed) does not change regardless on the curvature of the surface. Therefore, the connectivity of the open channel network to the surrounding is preserved regardless the location of pores on the surface. PCs have somewhat rounded facets on the surface compared to those of lipid cubosomes. However, we could not observe the specific defects specifically on the facets on the PCs. Please be aware that our imaging techniques are mostly based on electron microscopy, which is done under harsh conditions for PCs. We tried to use cryo-TEM for the tomography, but under our experimental conditions, we could not obtain enough images for constructing 3-D image due to the melting of ice caused by the flux of e-beam.

Other source of defects on the PCs could (1) disorder at the surface of the PCs caused by the interfacial curvature of the spherical PCs and the polycrystalline nature of the internal triply periodic minimal surface; (2) physical defects (broken membranes) at the surface of the PCs due to the high glass transition temperature (T_g) of the hydrophobic PS block. Although the disorder at the interfacial bilayers introduced deformation of lattices at the surface of the PCs, we found that the disorder at the interfacial bilayers and internal grain boundaries between crystalline domains of the minimal surfaces did not affect the non-interconnecting nature of two water-channel networks of the PCs. As previous results of the calculation of lipid cubic phases suggested, the closure of one channel network at the interfacial bilayer of PCs is a consequence of the self-assembly process to minimize the interfacial energy by not exposing the hydrophobic compartment of the bilayer to water.

Physical defects at the surface of PCs indeed break the non-interconnectedness of two channel networks. After self-assembly, the BCP bilayer composed of PCs becomes a glassy solid, which is prone to mechanical ruptures when exposed to external forces. By

SEM experiments, a small fraction of PCs showed the broken surface bilayers, which could result in the formation of double networks after replication. The population of double networks seemed to be increased when we rubbed the dry PCs to introduce more surface defects. Although we could not quantify the defected PCs, we believe that the possibility of having defects causing the interconnectedness of two channel networks should be very low.

(2) Figure 2d: z-sections: Why are the outer pores in the z-planes not color coded? Are they impossible to assign? And why is this the case?

[Our response]

→ Under our experimental condition for the TEM tomography, the penetration depth of electron beams (120 kV) was limited to less than ~ 200 nm. Also, the surface curvature of a spherical PC make the 3-D rendering of the TEM images of a PC very difficult. Therefore, we only chose a small section of a PC to addressing the locations and connectivity of two channel networks for the clarity of the image. For figure 2d, we intentionally chose 9 pores representing 2×2 lattices, and color-coded the surrounding pores to show the two channel networks are arranged in $Im\bar{3}m$ symmetry (figure 2d inset). The rest of pores could also be addressed as shown in the figure below, but we only chose a part of the channel networks for the sake of clarity.

(3) The vesicle fusion experiments are very clever. I enjoyed this section very much. However, the fluorescence profile in Figure 6g are somewhat misleading. The profile of the vesicle is diffraction limited. It implies a maximum in the center, but only because the flat profile cannot be resolved at higher magnifications if looking at the z-section. I would recommend to remove this. It is unclear what the authors wish to demonstrate with this?

[Our response]

→ We thank the reviewer for this comment and suggestion. Throughout our experiments, we observed a marked difference in the intensities of fluorescence from the unfused polymer vesicles and the PCs fused with vesicles. By considering the volume difference of the internal lumen of a polymer vesicle ($\sim 0.5 \mu\text{m}^3$ assuming the diameter of $1 \mu\text{m}$) and a closed channel network of a PC ($\sim 14.7 \mu\text{m}^3$ by assuming the volume fraction of channels of PCs having diameter of $5 \mu\text{m}$), we inferred that the diffusion of fluorescein from the polymer vesicle to a closed compartment of a PC would dilute the concentration of fluorescein, resulting in the difference in fluorescence intensities. Ideally, if a given number of fluorescein molecules are localized at the inner lumen of a vesicle (diameter $1 \mu\text{m}$), the same number of fluorescein molecules would be dispersed in a closed compartment of the PC (diameter of few μm). The intensity profiles were placed to demonstrate that fluorescein-encapsulating vesicles show higher intensity than that of the fused PCs. With reviewer's comment in mind, we removed this data from the figure 5 (revised manuscript) to avoid misleading.

(4) Control experiment: I a missing the following control experiment. Strictly speaking the fluorescence section does not proof that only one channel is filled, as the microscopy does not allow to resolve these details. Imagine there was a chemical interaction of what so ever origin

so that fluorescein segregates to the PEO or to the PEO-PS interface, then very similar images could be obtained, even for open pores. Hence, I recommend the following controls: (i) mixtures of prepared cubosomes and fluorescein, separation via sephadex and imaging. Ideally the cubosomes should not be enriched with the fluorophore to claim entrapment in closed pores. (ii) The authors should encapsulate one fluorophore in the closed pore network and add another fluorophore to infiltrate the open pores. The cubosomes with the larger lattice spacing should allow a SIM based imaging should allow for a separation of both dyes in the orthogonal channels.

[Our response]

→ The block copolymer was allowed to self-assemble into PCs in the presence of fluorescein in water. To show that the resulting PCs possess highly ordered cubic mesophases that are identical to PCs shown in figure 1, we added new data in the supplementary figure 5. After the self-assembly, we removed unencapsulated fluorescent molecules by dialysis against water and repeated centrifugations on a centrifugal filter (cut off 100 kDa) and exchange of a medium with distilled water (our method section). Size exclusion chromatography using Sephadex was not possible due to the large size of PCs. After these purification steps, the confocal laser microscopy images of the PCs at different focal planes clearly showed that fluorescein was well-retained within a closed compartment of the PC. (Supplementary Figure 5d). This indicates that the fluorescein is entrapped in a closed water-channel network without being diffused to the open channel network. In addition, SEM and SAXS analysis of the PCs self-assembled in the presence of fluorescein showed the presence of well-defined internal structures having bicontinuous water channel networks (Supplementary Fig 5h and i). For the control experiment, fluorescein was allowed to diffuse into the open channel network of the PCs formed without fluorescent molecules. We confirmed that the fluorescein diffused into an ‘open channel’ of PCs cannot be retained during the same purification processes. This result indicates that the retention of dye molecules (fluorescein sodium salt) by physically or chemically adsorption on the bilayer membrane could not be achieved. We added revised data in the supplementary figure 5.

→ **Modified Supplementary Figure 5**

Supplementary Figure 5. CLSM and SIM images of the PCs of PEG550₃-PS₁₅₀. **a,b,c**, CLSM images of PCs self-assembled in the presence of Fluorescein in water ($\lambda_{\text{Ex}} = 460 \text{ nm}$, $\lambda_{\text{Em}} = 515 \text{ nm}$) : (a) dark field, (b) bright field, (c) merged. The merged image showed the presence of dye in the water channels in PCs. **d**, SIM images obtained from different focal planes (interval of z-direction : 100 nm). Scale bars : 2 μm . **e,f,g**, CLSM images of PCs mixed with fluorescein in water after purification : (e) dark field, (f) bright field, (g) merged. The dark field and merged images did not show any retention of dye molecules. **h**, SEM images of fluorescein-encapsulating PCs. The inset shows magnified view of perforated surface of colloidal PC. **i**, SAXS result of fluorescein-encapsulating PCs shows that the phase and lattice constant are similar to PCs formed without fluorescein ($Im\bar{3}m$, $a = 60.2 \text{ nm}$).

→ We completely agree with the reviewer’s comment on the experiment with PCs with large lattice parameters. Our on-going experiments involve the synthesis of block copolymers using functionalized PEGs. Self-assembly of these block copolymers into PCs provides surface functional groups in a high density, which can covalently anchor fluorescent molecules on the surface of the open channel network of the PC. With the surface functionalized PCs, the visualization of two non-interconnecting channels of the PC should be possible by using super-resolution microscopy.

(5) The relation to biosynthesis of insect cuticles is to my opinion a bit overstated. Do we really learn from this processes that the formation of the cuticles works similarly?

[Our response]

→ There are two hypotheses on the mechanism for the synthesis of photonic crystals of insects using biological cubic membranes as the templates. One is the “prefolding” model: the intracellular SER membrane forms cubosome-like phases known as ordered smooth endoplasmic reticulum (OSER). From the folding of SER and plasma membranes, one of two non-intersecting water channel networks embedded within the OSER is connected to the extracellular space which allows the deposition of chitin. After the polymerization of chitin and cell death, only a single network structure is left as scales. The other is the

“cofolding” model in which chitin extrusion and membrane folding occur simultaneously. (ref. 6, *J. Morphol.* **202**, 69–88 (1989)).

In our study, we showed that the interfacial topology of the PCs could force only one of two non-intersecting water-channel networks to be exposed to the surroundings. This surface topology could be reversed upon the fusion with a polymer bilayer, resulting in the switch of the accessibility of the channel networks (i.e., open to closed, closed to open) at the interface created by fusion. We assumed that if the same process could happen in the epithelial cell of insects, the smooth endoplasmic reticulum (OSER) forms ordered particulates (cubosomes), which subsequently fuse with a plasma membrane. The fusion between the cell membrane and the cubosome (OSER) exposes the formerly closed channel network to the extracellular space, while the previously open channel network turned to a closed channel residing inside the cell. Chitin diffuses and polymerizes only in the exposed channel networks, and the cytosolic content remains within the cell. Essentially, this epithelial cell is diseased, and only chitinous materials remain as a scale. (The structure of the scale is hierarchical.).

Our experiments provide a model system for the ‘pre-folding’ model and a possible mechanism suggesting how the OSER could connect only one water-channel to the extracellular space without losing the cytosolic contents, which has not been suggested yet. With reviewer’s comment in mind, we revised our manuscript (P. 10 lines 228-235).

→ **Revised manuscript (P. 10 lines 228-235)**

The proposed mechanism for the synthesis of photonic crystals of insects using biological cubic membranes as the templates is the “prefolding” model: the intracellular SER membrane forms a cubosome-like phase (OSER). From the folding of SER and plasma membranes, one of two non-intersecting water channel networks embedded within the OSER is connected to the extracellular space which allows the deposition of chitin. After the polymerization of chitin and cell death, only a single network structure is left as scales⁶. Our suggested mechanism could complement the proposed mechanism (prefolding model) by suggesting the possible means to expose one water channel network to the extracellular space, which allowed the influx of chitin into the OSER template residing in the cell.

(6) Figure 5 and 6 should be merged.

[Our response]

→ We moved Figure 5 to the supplementary information as the supplementary figure 9. To this figure we also added the mechanism of membrane fusion to help readers.

→ **Modified Supplementary Figure 9**

Supplementary Figure 9. A schematic representation of the topological inversion at the interface between membranes. a, Interlamellar attachment (ILA). b, Polymersomes and PCs created by fusion. Polymer vesicles containing fluorescent molecules undergo fusion to the PC, resulting in the relocation of bilayer membranes to connect the lumen of a polymer vesicle to the closed channel embedded in a PC. Fluorescent molecules residing in the lumen of a polymer vesicle are confined within the closed channel of a PC because the inversion of the topology only occurs at the interface created by fusion. The open channel is closed at the interface.

(7) A recent minireview on cubosomes and similar structures should be cited “DOI: 10.1002/anie.201703765”

[Our response]

→ We thank for the suggestion. We added the reference (ref. 31).

REVIEWER #3

La and co-workers present a way to produce polymer cubosomes (PCs) and transform their internal structure into inorganic materials. The authors show that only one network of their PCs is connected to the environment and that this can be used to trap fluorescent molecules in the closed-off compartment or refill the dual network from the outside. While their results are certainly interesting and novel, some of the results need to be described better (and more honest) and to be placed into context. Let me detail these points below.

(1) Structure of the PCs? The authors have synthesized various PEG-PS polymers to form PCs with either a D or P minimal surface internal morphology. While the structure of the small PCs is convincingly either D/P, it is not clear how the channel morphology. However, it is not clear how the pore size (or its filling fraction) is related to the polymer morphology. Are the network directly connected to the PS/PEO fraction?

[Our response]

→ Our previous studies on the self-assembly of block copolymers in dilute solution showed that the block ratio, defined by the weight fraction of the PEG domain (f_{PEG}), is a deciding factor to determine the lattice of the bicontinuous cubic phase of a block copolymer (ref. 32, *Angew. Chem. Int. Ed.* **54**, 10483–10487 (2015)). In this study, we chose two block copolymers, PEG550₃-PS₁₅₀ and PEG550₃-PS₁₆₈, to demonstrate two distinct internal lattices of PCs and their interfacial topology. To show the internal channel morphologies, we compared 3-D reconstructed TEM tomograms and computer-generated projections of internal networks of PCs (Supplementary Figure 4). The pore size and internal volume of the PCs were previously measured by N₂ adsorption-desorption experiments (Ref. 30), which showed the type IV isotherms indicating that the interconnected porous networks are present in the PCs. The P surface generally have larger pore size than the D surface as the lattice parameter of the P surface is larger than that of the D surface while the thickness of the bilayer was identical for both minimal surfaces.

(2) Large PCs? What I fully disagree with is the structure determination of the large PEG2000-PS polymer of figure 4. Rather than wanting to ascribe this structure to a mixture of P/D, the results show that it is rather a disordered network of pores. The structure factors could also “just” be structure factors of pores and look quite similar to normal spheres. The SEM/TEM images do not convince me that the structure is indeed a single or a mix of morphologies. Also, the optical results underpin a more quasi-ordered network as the optics quite closely resembled the quasi-ordered network, e.g. found in birds (Tinbergen et al. JEB 2011 or Saranathan JRSI 2012). Please adjust accordingly.

[Our response]

→ We thank reviewer for raising this point. In the case of the PCs of PEG550₃-PS₁₅₀ and PEG550₃-PS₁₆₈, the order of the internal structure was very high partly because the block copolymers could reach the thermodynamic equilibrium during the self-assembly. Also, the presence of relatively short PEG chains ($M_n = 550 \text{ g mol}^{-1}$) in the hydrophilic corona of the bilayer membrane provides limited colloidal stability to the resulting PCs. Therefore, the average diameter of the PCs to be large ($> 11 \text{ }\mu\text{m}$) compared to the lattice constant of the bicontinuous cubic phases. These facts contributed to the size of the internal crystalline domains of a PC to be large enough for showing clear diffraction peaks by SAXS experiments. For these samples, we observed well-resolved peaks from

the SAXS experiments, which could be assigned to the corresponding lattices (Supplementary Figure 1).

However, for the PCs of PEG2000₃-PS₂₁₄₀, the mobility of the block copolymer during the self-assembly is limited due to substantially higher molecular weight of the PS block compared to the previous block copolymers. The presence of long PEG chains on the surface of the bilayer should increase colloidal stability of the PCs (ref. 32), which resulted in the formation of relatively small particles (average diameter = 2.7 μm). Due to the small diameter, the surface curvature of the PCs of PEG2000₃-PS₂₁₄₀ is much higher than the surface curvature of larger PCs of PEG550₃-PS₁₅₀ and PEG550₃-PS₁₆₈. This high surface curvature might result in the distortion of the lattices at the surface of the PCs. Also, the limited mobility of the block copolymer makes the self-assembled structures difficult to reach the equilibrium during the self-assembly, which contributes to the increase of disorder in the internal bicontinuous mesophases of the PCs. Also, the small diameter of the PCs of PEG2000₃-PS₂₁₄₀ gave a background noise from particle scattering in the SAXS experiments, which render the peaks to be broaden significantly as shown in Figure 4d.

Arising from the disorder in the mesophases, SiO₂ and TiO₂ replica of the PCs showed broad peaks by SAXS experiments, which could be assigned to $Fd\bar{3}m$ symmetry as shown in Supplementary Figure 7. From the SEM and TEM studies of these skeletal replica of the PCs, we observed that the lattices of the skeletal structures consisted mostly of a single diamond network ($Fd\bar{3}m$ symmetry, four-fold junctions, Figure 4e inset and 4h). However, we also observed the presence of primitive cubic networks from the surface of the skeletal replica, which might arise from the distortion of the lattice at the surface of the PCs. Based on the analysis of SEM images, we inferred that a minor fraction of the networks had a primitive cubic ($Pm\bar{3}m$) structures, which mostly appeared at the surface of spherical particles composed of single networks (supplementary figure 8). Based on these observations and our previous results of the PCs having a mixed phase (*Angew. Chem. Int. Ed.* **54**, 10483–10487 (2015); *ACS Nano* **9**, 3084–3096 (2015)), we assigned our SAXS results to the mixed phases. However, due to the less resolved and broad SAXS peaks, we could not confirm the coexistence of two phases within the PCs and replica particles. Therefore, we reassigned our SAXS data of replicas to $Fd\bar{3}m$ symmetry with noticeable disorder in the structures. The disorder of the lattices of the PC, however, did not disrupt the interfacial topology, resulting in the formation of single network structures. We revised our manuscript accordingly (P. 8 lines 180-182, 183-186, P.9 lines 189-197) and added revised data in the supplementary information.

- We also compared the reflectance profile measured from the feathers of bird with our replicas. However, despite the structural differences, reflectance spectra did not reveal significant difference in bird feathers, butterfly wing scales and our replicas (ref. 3, 4, and *J. R. Soc. Interface* doi:10.1098/rsif.2012.0191). Unlike the secondary peak of reflectance profile of bird feathers, our TiO₂ replicas show the primary peak at 525 nm due to the photonic structures and the peak near 400 nm is TiO₂'s intrinsic absorption (~380 nm wavelength). In our case, all the samples of TiO₂ replicas exhibited results similar to the reflectance spectra, which could be found from TiO₂-based photonic crystals (*Environ. Sci. Technol.* **44**, 451–455 (2010), *J. Alloys compd* **769**, 740–757 (2018)). We added more explanations about reflectance spectroscopy of TiO₂ (P.9 line 203)

UV-vis. Reflectance spectrometry of the TiO₂ replicas showing TiO₂ absorption edge at 380 nm.

→ **Revised manuscript (P. 8 lines 180-182)**

The diameter of PCs is relatively small ($d = 2.7 \mu\text{m}$), indicating the increased colloidal stability arising from the presence of long PEG chains on the surface of the bilayers³².

→ **Revised manuscript (P. 8 lines 183-186)**

The SAXS results of the PC showed broad peaks suggesting significant distortion of the lattice in the internal cubic mesophases of the PCs (Supplementary Fig 7). From the primary peak of the SAXS result, the lattice parameter was estimated to 213 nm by assuming that the internal Schwarz D surface ($Pn\bar{3}m$ space group).

→ **Revised manuscript (P. 9 lines 189-197)**

The resulting skeletal frameworks were characterized by SEM, TEM, and SAXS, which indicated the presence of cubic crystalline single network (Fig. 4d–f, h, Supplementary Fig. 7). SAXS and SEM images of the SiO₂ and TiO₂ replica of the PC of PEG2000₃-PS₂₁₄₀ suggested that the lattice of the single cubic network was mainly retain a single diamond structure while significant distortion of the lattice was present in the skeletal network (Supplementary Fig. 8). We tried to assign the SAXS results obtained from these skeletal networks, which was unsuccessful due to significant broadness of the peaks (Supplementary Fig. 7g). In spite of distortion of the lattices partly caused by the high molecular weight of the BCP and the high curvature at the surface of the PCs, the replicated SiO₂ and TiO₂ structures showed a single cubic network having nodes of 4-fold symmetry.

→ **Revised manuscript (P. 9 lines 203)**

The peak near 400 nm indicates an intrinsic absorption edge of TiO₂

→ **Modified Supplementary Figure 7**

Supplementary Figure 7. Structural characterization of the titania replica of PCs of PEG2000₃-PS₂₁₄₀. **a**, Low-magnification SEM image showing spherical replicas. **b,c**, SEM images of structures of the TiO₂ replicas showing $Pm\bar{3}m$ space group (b) and $Fd\bar{3}m$ space group (c). **d,e**, TEM images showing the single network of replicas. **f**, Normalized SAXS results of skeletal silica replicas (red line) and titania replicas (blue line). Vertical lines correspond to the expected Bragg peak positional ratios of $Fd\bar{3}m$ space group. **g**, SAXS results of the PCs and replicas assigned to mixed phases. The expected peak position of $Im\bar{3}m$ space group of PCs was calculated from the lattice parameter relationship of two phases ($a_{p\ surface}/a_{d\ surface} = 1.279$). Polymer cubosomes; $Pn\bar{3}m$ ($a = 213$ nm, black) and $Im\bar{3}m$ ($a = 273$ nm, red). SiO₂ replicas; $Fd\bar{3}m$ ($a = 361$ nm, red) and $Pm\bar{3}m$ ($a = 231$ nm, black). TiO₂ replicas; $Fd\bar{3}m$ ($a = 247$ nm, blue) and $Pm\bar{3}m$ ($a = 159$ nm, black).

➔ Added Supplementary Figure 8

Supplementary Figure 8. SEM images of the SiO₂ replica of PCs of PEG2000₃-PS₂₁₄₀. **a,b**, Low-magnification SEM image showing the single cubic network of the spherical replica (a) and magnified view of surfaces of the replica showing distorted lattice (b). **c**, SEM image of the single cubic network of SiO₂ replicas showing $Pm\bar{3}m$ space group on the surfaces.

(3) Why is only one network connected to the outside? While the results of Figures 1-4 are really beautiful, the results and discussion of figure 5-6 are purely hypothetical. I would argue that Figure 5 is really clear and trivial and should go either to the SI or out, as it does not really make sense to me. Please make sure that this is pure speculation that it is connected to the growth of SERs as this is not convincingly shown, as you also do not discuss as to why only one channel of your PCs is connected. Please amend.

[Our response]

→ We thank reviewer for the comments. With the suggestion in mind, we moved figure 5 to the Supplementary information (**Supplementary figure 9**) following the reviewer's suggestion. To this figure we also added the mechanism of membrane fusion to help readers.

→ **Modified Supplementary Figure 9**

Supplementary Figure 9. A schematic representation of the topological inversion at the interface between membranes. a, Interlamellar attachment (ILA). **b,** Polymersomes and PCs created by fusion. Polymer vesicles containing fluorescent molecules undergo fusion to the PC, resulting in the relocation of bilayer membranes to connect the lumen of a polymer vesicle to the closed channel embedded in a PC. Fluorescent molecules residing in the lumen of a polymer vesicle are confined within the closed channel of a PC because the inversion of the topology only occurs at the interface created by fusion. The open channel is closed at the interface.

→ Figure 5 and the related paragraphs in the revised manuscript describe our experiment to show how the PC can encapsulate guest molecules in one channel network. We showed that the diffusion of free fluorescein molecules into the open channel of the pre-formed PCs did not result in the retention of dye molecules in the water channel network of PCs (**P. 6 line 133-135**). We realized that polymer vesicles could be fused in the presence of

organic solvents in an aqueous medium (see our method section and the figure below). Considering the mechanism of the membrane fusion (Supplementary figure 9), we devised a way to introduce guest molecules to the closed channel of the PC by the fusion of a polymer vesicle and a PC. As the mechanism of the fusion of bilayer membranes suggests, the fusion between a vesicle and a PC would result in the inversion of topology at the newly created membrane by fusion (Supplementary figure 9a). This fusion connects the aqueous lumen of the vesicle and the closed compartment of the PC, which results in the diffusion of fluorescein molecules encapsulated within the vesicle. We found the encapsulation of fluorescein molecules within the PC fused with a vesicle. We believe that our results suggest that how the OSER template residing within a cell could be exposed to the extracellular space, and how only one water channel of the OSER could be connected to the extracellular space. Our results complement the existing prefolding mechanism for the synthesis of biophotonic crystals in insects. With reviewer's comment in mind, we revised our manuscript (P. 10 lines 217-219, 228-235).

→ **Revised manuscript (P. 10 lines 217-219)**

In this condition, polymersomes of PEG550₃-PS₁₂₀ could undergo fusion although the population of the fused polymersomes was small (Supplementary figure 10d).

→ **Modified Supplementary Figure 10**

Supplementary Figure 10. Fluorescein-encapsulating polymersomes and their fusion with PCs. a,b, Size distributions (a) and TEM image (b) of the fluorescein-encapsulating

polymersomes of PEG550₃-PS₁₂₀ ($d = 992$ nm). **c**, CLSM image of fluorescein-encapsulating polymersomes. **d**, CLSM images showing the fusion of fluorescein- and rhodamine B-encapsulating polymersomes. White arrows indicate fused polymersomes which exhibit orange color in merged images. **e**, SIM images of a fused PC obtained from different focal planes (interval of z direction : 800 nm).

→ **Revised manuscript (P. 10 lines 228-235)**

The proposed mechanism for the synthesis of photonic crystals of insects using biological cubic membranes as the templates is the “prefolding” model: the intracellular SER membrane forms a cubosome-like phase (OSER). From the folding of SER and plasma membranes, one of two non-intersecting water channel networks embedded within the OSER is connected to the extracellular space which allows the deposition of chitin. After the polymerization of chitin and cell death, only a single network structure is left as scales⁶. Our suggested mechanism could complement the proposed mechanism (prefolding model) by suggesting the possible means to expose one water channel network to the extracellular space, which allowed the influx of chitin into the OSER template residing in the cell.

(4) Figure layout. I would strongly suggest to group the results for the different PCs in the figures, e.g. move all belonging to Schwarz D to the right and the one's to P to the left. This will make it easier to compare. Also, many figures are missing scale bars (Fig. 1 G-J, Fig. 2 i). It also not clear which colored curve in Figure 3 belongs to which sample.

[Our response]

→ We thank reviewer for this suggestion. We modified the layout of figure 1 and added scale bars in the figures. In Figure 3, we re-color coded the SAXS data and EM images.

(5) Language. Please have the language checked again as some “the” are too many, some “the” are missing. Furthermore. “single networks” of...? In particular, the summary needs some work as “obscuring” and “to become” are more teleological and scientifically correct. The summary could also profit from a bit more detail.

[Our response]

→ We appreciate the invaluable comment. As the referee requested, we have revised the manuscript.

(6) Methods. While the PC process is clear, the process has no details on the spectroscopic characterization, the SAXS measurements or the details on other instruments. Please add to allow reproducibility.

[Our response]

→ As the referee commented, we found that one page was missing from the supplementary information about methods section. We corrected this in our new supplementary information. We apologize about this mistake.

Detailed comments.

P. 2: define “low coordinate”

→ We revised manuscript.

→ **Revised manuscript (P. 2 lines 35-37)**

The biosynthesis of single networks of open-space cubic lattices such as simple cubic, single gyroid, and single diamond symmetries.

P. 4: don't start with “in order to test our idea”, quite jargon

→ We changed the sentence from “in order to test our idea” to “In this study” (P. 4 line 74)

P. 5: the 3D reconstruction description needs to be improved.

→ We added the description.

→ **Revised manuscript (P.5 lines 112-117)**

The PCs were taken at different tilt angles, offering successive recording of two-dimensionally (2-D) projections from different views. The tilt series of images were carefully aligned and used to reconstruct the entire volume of PC with nanometer-scale spatial resolution using back-projection algorithms in IMOD. The three-dimensionally (3-D) reconstructed volume image of the PC of PEG550₃-PS₁₅₀, shown in Fig. 2a,

→ **Revised manuscript (P.6 lines 124-125)**

The cross-sectional views (with each intervals of 12 nm) of the TEM tomograms

P. 6: add note on volume fractions of the channels here.

→ For the lipid bicontinuous cubic phases, the calculation of volume fractions of the lipid phases was demonstrated by Tuner et al. (*J. Phys. II* **2**, 2039–2063 (1992)). Based on this estimation, the calculated volume fractions of channels of the PCs are 0.548 ($Im\bar{3}m$) and 0.599 ($Pn\bar{3}m$). In addition, the calculated pore diameters of the PCs are 11 nm ($Im\bar{3}m$) and 10 nm ($Pn\bar{3}m$). Experimentally, these values are off from our measurements by N₂ adsorption-desorption experiments to estimate the surface area and internal pore diameter and pore volume of PCs (ref. 30,32,38). The average pore diameters of the PCs were 20~30 nm.

N_2 adsorption-desorption isotherm (left) of the PCs of PEG550₃-PS₁₅₀ showing Type IV isotherms with type H2 hysteresis. A Brunauer–Emmett–Teller (BET) surface area was 57.4 m² g⁻¹ and a pore volume was 0.38 cm³ g⁻¹. The Barret-Joyner-Halenda (BJH) pore-size distribution curve (right) showed a broad range of pores with an average value of 28 nm.

P. 7: what is black/red in the curves? Add description here.

- We added detailed description to the figure legend (Figure 3). Vertical lines correspond to the expected Bragg peak positional ratios (magenta and red; common peaks, dashed red; $Fd\bar{3}m$ only, black; PCs only).

L. 163: a recent paper <https://onlinelibrary.wiley.com/doi/abs/10.1002/smll.201802328> highlights this in quite some detail. Maybe add a note to this ms here?

- We added reference (ref. 37) and revised the manuscript.
- **Revised manuscript (P. 7-8 lines 163-168)**
Recently, Akbar and coworkers reported the platinum nanowire networks having a single diamond symmetry with a lattice parameter of 13.2 nm via electrodeposition of metal precursors by using a double diamond cubic phase of lipids as a template³⁶. Our results indicated that the PC composed of BCPs possessed the identical structures to those of lipid cubic mesophases and could serve as templates for the synthesis of single cubic networks with larger lattice parameters³⁷.

L. 175 and elsewhere: give errors where possible.

- It is difficult to display error because of large dispersity. $d = 2.7 \mu\text{m}$ (1.5~ 5,8 μm)

L. 190-195: as described above the optics is more similar to quasi-ordered networks than to a mix of two unit cells, please adjust.

- We revised our manuscript (P. 8 lines 180-186, P. 9 lines 189-197, 203).

L. 208: the description of Figure 6 needs to be improved. Not really clear.

→ We modified description about the fusion experiment and the figure.

→ **Revised manuscript (P.10 lines 219-220).**

After removal of excess dyes, we observed both unfused free polymersomes and fused PCs showing the presence of encapsulated fluorescein (Fig. 5a and b).

L. 221: what is exactly the process of behind the fact that only one network is connected? Only with that you can draw further conclusions.

→ As mentioned in the introduction, cubosomes composed of TPMSs of lipid would adopt a topology in which one channel network is sealed while the other remains open the surroundings. In addition, Akbar and coworkers (*Adv. Mat.* **25**, 1160-1164 (2013)) reported the synthesis of 3D platinum nanowire networks with single diamond morphology by using thin layer of a self-assembled double diamond phase of lipid. With reviewer's comment in mind, we revised introduction part (P. 3 lines 54-56). We believe that the connectivity of one channel of double networks of PCs was well explained with various experiments like encapsulation, replication, and fusion.

→ **Revised manuscript (P.3 lines 54-46)**

This topology of the circumferential bilayer presumably arises due to the minimization of the interfacial energy by not revealing the hydrophobic compartment of the bilayer to the aqueous medium.

L. 233: how can a structural color be reminiscent of iridescent of single networks? Quite a stretch and linguistically wrong. Please rewrite.

→ We rewrote the expression.

→ **Revised manuscript (P. 10 line 251-252)**

which display structural colors analogous to the iridescence exhibited by biophotonic crystals of insects

REVIEWERS' COMMENTS:

Reviewer #1 (Remarks to the Author):

Authors have carefully read reviewer comments and answers are convincing and appropriate. Accordingly, changes in the manuscript were done. Those, in general, improved the paper except the part on the experiments aiming at demonstrating that one channel is close and the other one is open.

This refers to p.6 lines 128 to 136.

Lines 129-133 are convincing and demonstrate that at least part of the water inside the particles are isolated from the surrounded media.

Lines 133-135 are not so clear. Authors should explain that first unloaded PCs were formed. Then Fluorescein was added. Finally, dialysis was performed as before and no trace of dye remains.

It should also be added that open pores are present as demonstrated by SEM (supplementary Fig. 5h) (as explained in the answer to reviewers).

Finally, it could be added that the tomography work presented before and the study on replication that will be presented in the next paragraph confirm the presence of one open and one close water channel.

Despite this minor point, article is of very good quality. It has improved compared to the first version and article deserves to be published in Nature Communications

Reviewer #2 (Remarks to the Author):

The authors have conducted a very thorough revision of their article, and the article is ready for publication. I believe that we are looking at an extremely insightful article discussing a very complex behavior of such highly structured block copolymer nanoparticles.

Reviewer #3 (Remarks to the Author):

The authors have nicely incorporated the suggested changes and the manuscript reads much better now. Nice work!

While all of my scientific comments have been adequately answered, I have to urge the authors again to give errors to their values, where it is possible. Measurement errors are the basis of scientific measurements and a measurement is never perfect. Particle size cannot be an excuse to not perform an error analysis.

We thank all reviewers for comments. We accepted reviewers' comments and revised our manuscript accordingly. The additions and corrections related reviewers' comments are highlighted with a yellow marker in the revised manuscript.

REVIEWER #1

Authors have carefully read reviewer comments and answers are convincing and appropriate. Accordingly, changes in the manuscript were done. Those, in general, improved the paper except the part on the experiments aiming at demonstrating that one channel is close and the other one is open.

This refers to p.6 lines 128 to 136.

Lines 129-133 are convincing and demonstrate that at least part of the water inside the particles are isolated from the surrounded media.

Lines 133-135 are not so clear. Authors should explain that first unloaded PCs were formed. Then Fluorescein was added. Finally, dialysis was performed as before and no trace of dye remains.

It should also be added that open pores are present as demonstrated by SEM (supplementary Fig. 5h) (as explained in the answer to reviewers).

Finally, it could be added that the tomography work presented before and the study on replication that will be presented in the next paragraph confirm the presence of one open and one close water channel.

Despite this minor point, article is of very good quality. It has improved compared to the first version and article deserves to be published in Nature Communications

[Our response]

→ As the referee suggested, we revised manuscript about the control experiment for the clear understanding. We also added the demonstration about the presence of open and close channels of PCs.

→ Revised manuscript (P. 6-7 lines 136-141)

SEM and SAXS analysis of the PCs self-assembled in the presence of fluorescein showed the same surface topology and well-defined internal structures of double networks (Supplementary Fig. 5h and i). These results indicate that the fluorescein is entrapped in a closed water channel network without being diffused to the open channel network of PCs connected to the surrounding medium. For the control experiment, we mixed pre-formed PCs with a fluorescein solution.

REVIEWER #2

The authors have conducted a very thorough revision of their article, and the article is ready for publication. I believe that we are looking at an extremely insightful article discussing a very complex behavior of such highly structured block copolymer nanoparticles.

REVIEWER #3

The authors have nicely incorporated the suggested changes and the manuscript reads much better now. Nice work!

While all of my scientific comments have been adequately answered, I have to urge the authors again to give errors to their values, where it is possible. Measurement errors are the basis of scientific measurements and a measurement is never perfect. Particle size cannot be an excuse to not perform an error analysis.

[Our response]

- ➔ We corrected the size of PCs with range of values. The particle size of PCs was measured by analyzing SEM images of PCs. One hundred particles were selected for the image analysis from low-magnified SEM images.
- ➔ **Revised manuscript (P. 4 lines 83-84)**
(average diameter (d) determined by analysis of SEM images = $11.1 \pm 6.7 \mu\text{m}$. \pm indicates range of values)
- ➔ **Revised manuscript (P. 4 lines 86)**
($d = 15.9 \pm 7.9\mu\text{m}$)
- ➔ **Revised manuscript (P. 4 lines 189-190)**
($d = 2.7 \pm 2.8\mu\text{m}$),